# Future changes in severe hail across Europe, including regional emergence of warm-type thunderstorms

Abdullah Kahraman [1,2,3] ✉, Elizabeth J. Kendon [4,5], Hayley J. Fowler[1,3] & Chris J. Short [4]

Hail is a major threat to agriculture, properties, and people, yet little is known about changes to hail with anthropogenic warming. Here, we use pan-European convection-permitting simulations, and a contemporary hail proxy benefiting from simulated thunderstorm features, and show that the potential for severe hail decreases under RCP8.5, except potentially for very large hail. This is despite an increase in the number of convective storms producing many small ice particles functioning as hail embryos. The decrease in severe hail potential is partly due to hail forming at higher altitudes as the atmosphere warms, which impacts both the updraft strength in the hail growth layer and the extent to which hail melts before reaching the surface. Our results contradict those from coarser resolution models which typically project future increases in hail frequency, estimated using environmental proxies. However, we find that future warm seasons feature a warmer thunderstorm type akin to hail-producing storms found in the tropics, where the largest hailstones can still reach the surface as evidenced from observations. In the future, these storms are most frequent over southern Europe, leading to regional increases in severe hail frequency. We conclude that society may need to be prepared for (infrequent but) more impactful hail in a future warmer world.

Hail is among the costliest of local hazards stemming from thunderstorms, yet despite this, there remain uncertainties about how it forms[1], challenges around its short-range forecasting[2,3] and its projection with a changing climate[4]. Recent developments in understanding hail environments[5-9] and the representation of relevant processes in convection-permitting climate models (CPMs)[10], coupled with improved computational capabilities, provide an opportunity to assess future hail threat at a continental-scale.

The severity of hail is mostly defined by its size, although the kinetic energy of hailstones also depends on the accompanying winds[11]. The size determines the terminal velocity that a hailstone can reach. In Europe, a 2 cm diameter is used as a threshold for large (or "severe") hail, and 5 cm and larger is considered as very large (or "significant severe") hail[12,13].

To form, hail requires an embryo, usually graupel (a small ice particle) or frozen droplets[14]. When the embryo is swept through a thunderstorm's updraught, it gets larger in size via accretion and/or riming, i.e., collecting supercooled water droplets (liquid water at temperatures below 0 °C), which immediately or gradually freeze onto its surface[15,16]. This process usually takes place in a layer from −10 to −30 °C. It is thus important how long the embryo stays within this 'hail growth zone' (HGZ), with the balance of updraught strength and hail size (which controls fall speed) crucial for maintaining growth. The most favourable environment for large hailstones is the mesocyclone (rotating updraught) of supercells, permitting a hailstone to stay longer in the HGZ, as it moves towards the inner (and hence stronger) part of the updraught in a curved, arching path. Significant severe hail

[1]School of Engineering, Newcastle University, Newcastle upon Tyne, UK. [2]Visiting scientist at Met Office Hadley Centre, Exeter, UK. [3]Tyndall Centre for Climate Change Research, Newcastle University, Newcastle upon Tyne, UK. [4]Met Office Hadley Centre, Exeter, UK. [5]Bristol University, Bristol, UK. ✉ e-mail: abdullah.kahraman@newcastle.ac.uk

almost exclusively occurs within supercells[6,17,18]. Once the hailstones fall below the zero-degree wet-bulb level they start to melt, with the largest hailstones having the greatest chance of surviving to reach the ground due to their lower surface area/mass ratio and higher fall speed[19–22]. Increases in melting height are therefore an important factor controlling changes in the size distribution and frequency of hail reaching the surface in a warming climate[23].

Previous studies exploring future changes in hail have estimated changes using favourable environmental conditions extracted from coarse-resolution climate models[24,25]. The advent of CPMs, however, allows a more explicit representation of hail and graupel[26]. One study for the contiguous U.S. (CONUS) used the WRF-ARW model at convection-permitting resolution, to assess future changes in hail during spring and summer[27]. The hail metric was derived from maximum column-integrated graupel, showing mostly a decrease in ≥2 cm hail in June and July, but increases in larger sizes across CONUS from March to August. The limitations of this approach were the use of graupel as a direct proxy for hail, a bulk microphysics scheme for which explicit size information is not available, and that the model resolution was too coarse to depict updraught/hail growth processes accurately. Another CPM study covering most of CONUS, found regional and seasonal differences in future changes in the number of storms reaching a reflectivity value of 60 dBZ[28] (which could be indicative of hail), with mainly increases in the eastern U.S. and decreases over the Great Plains. However, their approach also had limitations, such as model biases in reflectivity and the fact that they did not use a direct measure of hail. To date, there have been no pan-continental CPM studies for Europe that examine projected changes in severe hail using a physically-based hail diagnostic approach.

In this study, we use a process-based metric for hail (the Severe Hail Potential (SHP)[29]) that exploits the convective-scale outputs from CPMs. Using a pan-European CPM with a grid spacing of 2.2 km, where the updraughts and downdrafts of thunderstorms are produced by the model dynamics, we extract graupel amount, simulated vertical velocities within the HGZ, environmental winds, and thermodynamic profiles to calculate and apply a severe hail proxy. We examine differences in this proxy between 10-year future (for two different periods, 2040 s and ~2100, under the RCP8.5 emissions scenario) and 10-year control (1998–2007) simulations to assess the changing hail threat over Europe and identify the processes driving these changes.

## Results

### Change in the frequency of storms with high graupel content
Graupel (heavily rimed ice particles, of sizes up to 5 mm) production is a key feature of thunderstorms, and graupel data can be extracted from CPM simulations to assess individual thunderstorms[27]. Here, we first calculate graupel water path for each model time step, and then analyse their daily maxima (GWPmax). We find that, in the current climate, days with daily maximum graupel water path exceeding 10 kg m$^{-2}$ (Graupel10, hereafter) are most frequent in autumn over the Mediterranean Sea, whilst land areas experience these storms throughout the warm season (Fig. 1). This pattern follows the convection and lightning climatology of Europe [30]. In the future simulation (ca. 2100), high-graupel storms increase in frequency over the Mediterranean (and the surrounding coasts) in autumn and winter, and in summer over Northern Europe, including part of the British Isles, parts of Central Europe, as well as most of the Adriatic coast, with a slight increase in Southern Europe (SEU) in spring (Fig. 1). Large decreases are projected in the summer over the seas surrounding Italy (Fig. 1h), despite the increases projected over Italy. Notably, decreases are projected over Southwest Europe in summer and in Central Europe in spring and summer. In total, except for a narrow latitudinal band around Central Europe and local hotspots in the south, positive signals dominate the European domain (Fig. 1n). For instance, Southern Scandinavia currently experiences fewer than 0.02 high-graupel

storms per km$^2$ y$^{-1}$; this quadruples by the end of the century. The central Mediterranean and surrounding coasts show even greater increases. For storms with even higher graupel content (50 kg m$^{-2}$, as used in the literature[27] for very large hail, termed here Graupel50), future frequency increases are more confined to autumn and the Mediterranean region, with little difference seen elsewhere (Fig. 1l, o).

### Severe Hail Potential and its future change
Although there is an overall increase in storms with high graupel content, graupel alone does not necessarily translate into hail, especially severe hail (convective precipitation in the forms of balls or irregular lumps of ice, ≥2 cm in diameter). Many storms produce graupel without hail, while some with high graupel content produce large accumulations of small hail[31]. Indeed, in the U.S. simulations[27], the number of Graupel10 gridpoints was approximately twice as many as those with a maximum hail diameter of 2 cm (or larger), as computed by the HAILCAST model[3]. Thus, to constrain our analysis, we now diagnose simulated updraughts favourable for severe hail. Using instantaneous values with 3 h intervals, we examine occasions of 10 m s$^{-1}$ or higher HGZ vertical velocities (in grids where graupel water path is 100 g m$^{-2}$ or higher) within 10 m s$^{-1}$ or higher deep layer shear environments (indicative of more organised storms and potentially wider updraughts, supporting sufficient time for hail growth to large sizes, as described in the Methods and detailed in the study[29]). We exclude cases with freezing level height at ≤2 and ≥4.5 km above the surface to focus on the optimal conditions for hail: if the freezing level is too low, low-level temperatures are too cold for sufficient moisture; if the freezing level is too high, hail that forms will melt before reaching the surface (except perhaps the largest hailstones). We define SHP as when all the above conditions are met simultaneously at a grid point. Further, we extract a subset of the top 3% of severe hail cases to define "Significant Severe Hail Potential" (SSHP); these are cases with values of HGZ vertical velocity x deep layer shear exceeding 400 m$^2$ s$^{-2}$. The development of the SHP and significant SHP proxies and their validation against observed datasets is further detailed in the earlier study[29].

The overall spatial, seasonal, and diurnal patterns of severe hailstorms are well captured by our proxy[29]. The diurnal distribution of hail peaks in the afternoon during spring and summer, when convection is mainly dependent upon solar radiation over land areas. There is smoother diurnal variation in autumn and winter when Mediterranean storms dominate (Supp. Fig. 1). As observations are quite limited, we compared our proxy[29] to observations mainly for Central Europe (Supp. Fig. 2a,b) due to their reliability. The European Severe Weather Database (ESWD)[12,13] contains 6300 cases of hail with a maximum diameter of ≥2 cm in Central Europe between 2012 and 2021 (630 cases y$^{-1}$ over the 10 years, but with 1171.7 cases y$^{-1}$ in the last 3 years of record), while our control simulation projects an average of 1081.8 cases y$^{-1}$ with SHP. Given the well-known under-reporting of severe weather, especially in earlier years[1,13,32,33]), these values are comparable (Supp. Fig. 2b). However, we note that our severe hail proxy will underestimate probable severe hail cases in the CPM simulation, as the time interval between instantaneous data stored is 3 hours, which implies there may be missing cases in between. Further evaluation of the proxy, including its seasonal and spatial distribution across Europe (calculated with a 20-year hindcast simulation), is discussed[29], and shows good agreement with observations for most recent years and is an advance over existing climatologies.

We find that in the current climate, the potential for severe hail in continental Europe mainly arises in summer (Fig. 2e), and to a lesser extent, spring (Fig. 2c). Autumn is the peak month for hail over the Mediterranean Sea and coastal areas. Overall, the seasonal distribution of SHP in the control simulation agrees with the observations (Supp. Fig. 2) and the hindcast[29], except for higher values in spring. The spatial patterns roughly follow those of Graupel10, as expected, but the frequency values are lower by an order of magnitude, which is in line with

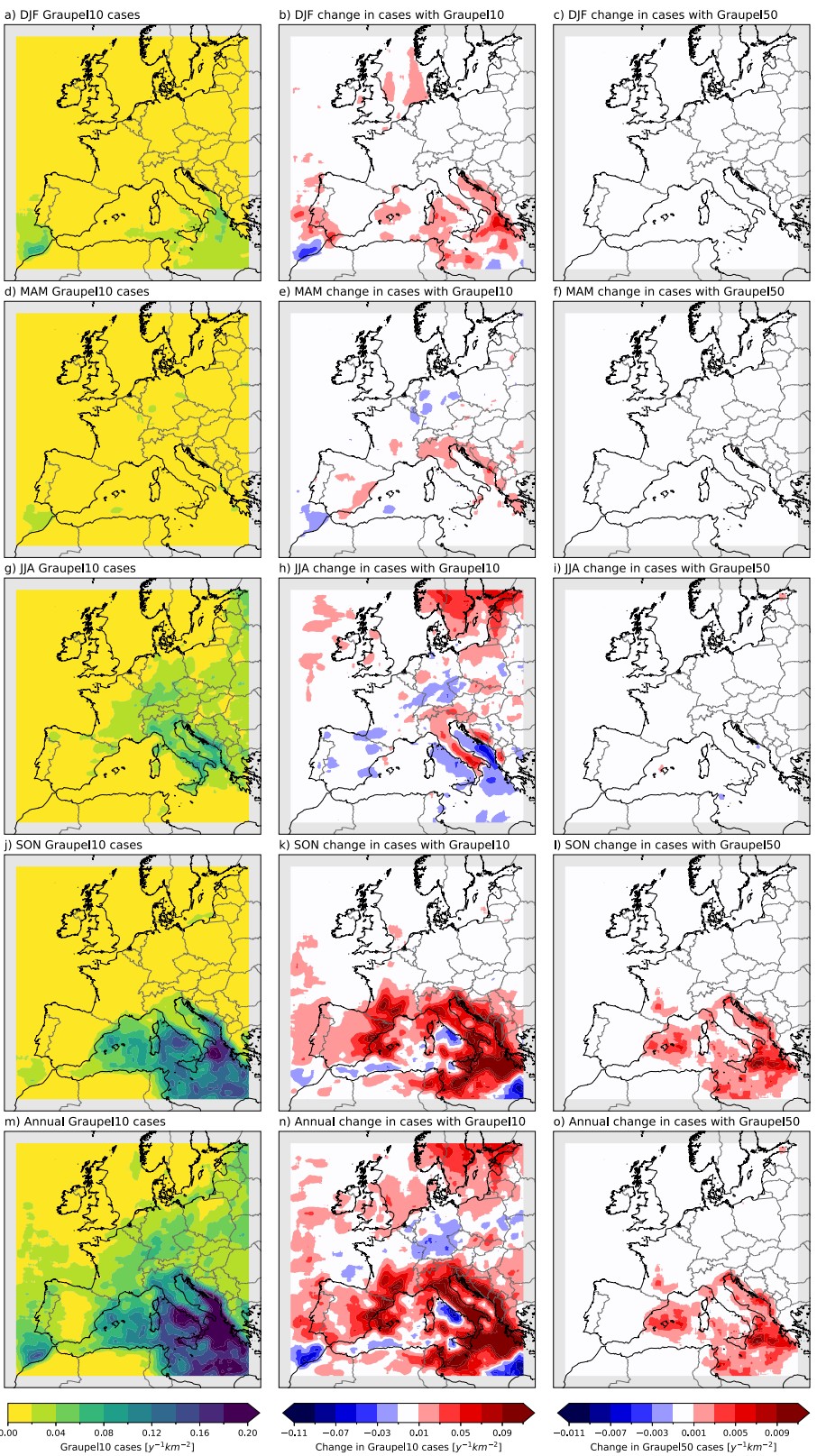

**Fig. 1 | Graupel-based analysis.** Number of cases with daily maximum graupel water path (GWPmax) exceeding 10 kg m⁻² per year per km² (Graupel10) in the historical period, future (end-of-century) changes in Graupel10, and future (end-of-century) changes in Graupel50 (as in Graupel 10, but with 50 kg m⁻² threshold): **a–c** for December, January and February (DJF), **d–f** for March, April and May (MAM), **g–l** for June, July and August (JJA), **j–l** for September, October and November (SON), **m–o** for whole year, respectively. The GWPmax is calculated for each time step. The current climate corresponds to 1998–2007, and future climate is a comparable 10-year simulation corresponding to year 2100 under RCP8.5. A smoothing has been applied by averaging the neighbouring ±25 grid points. 70 grids from each lateral boundary are excluded from the analysis to remove boundary artefacts (grey shading). Note that the scale of Graupel50 plots is one order of magnitude smaller than that of Graupel10.

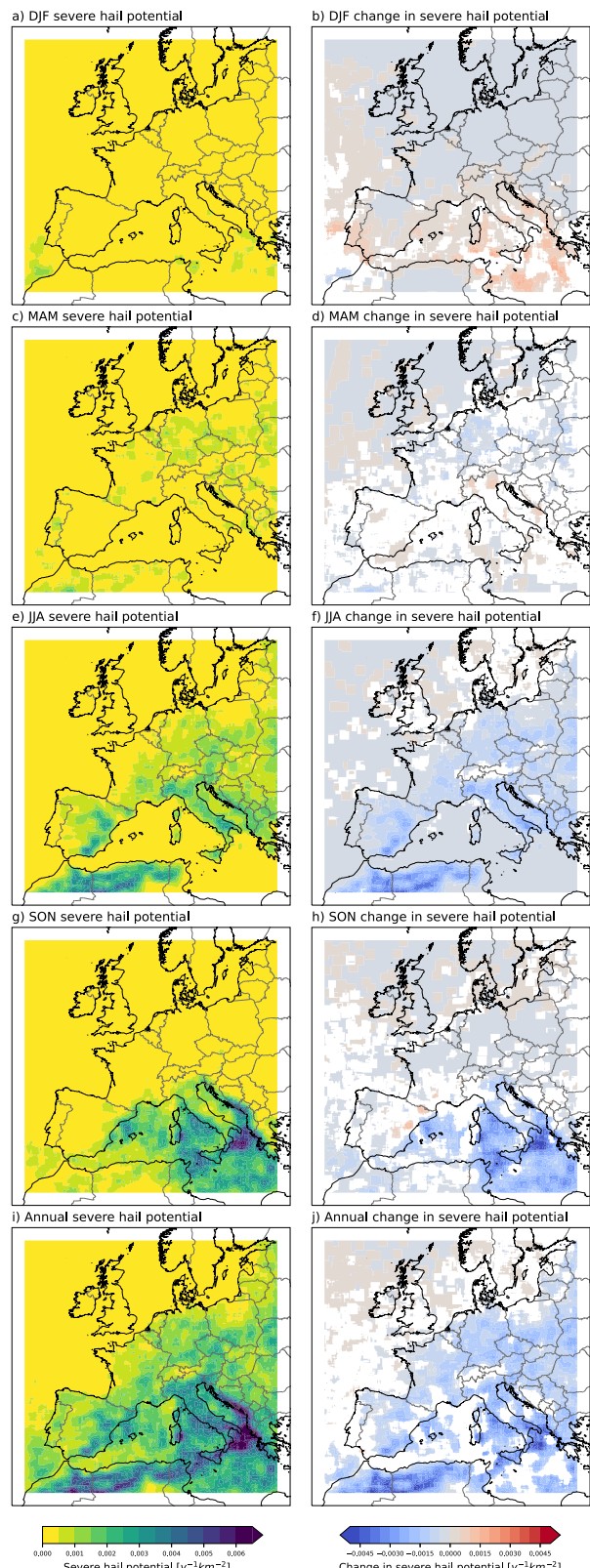

**Fig. 2 | Severe hail potential and its change.** Severe hail potential in the current climate, and future (end-of-century) changes in severe hail potential, for December, January and February (DJF; **a**, and **b**, respectively), for March, April and May (MAM; **c**, **d**), for June, July and August (JJA; **e**, **f**), for September, October and November (SON; **g**, **h**) and for whole year (**i**, **j**). The current climate corresponds to 1998–2007, and future climate is a comparable 10-year simulation corresponding to year 2100 under RCP8.5. A smoothing has been applied by averaging the neighbouring ±25 grid points. 70 grids from each lateral boundary are excluded from the analysis to remove boundary artefacts. For the future change maps, only significant changes at the 10 percent level are shown, based on 1000 bootstraps. Grids with insignificant changes are masked in white.

thermodynamic profile, and the hydrometeor type of severe storms, rather than a change in the frequency of severe storms. It is well known that higher melting levels and larger warm cloud depths with high liquid water content will more likely produce heavy rain, rather than severe hail[23,34]. For instance, as shown by earlier studies, our future simulation has much higher melting level height[30], higher moisture availability[35], and more frequent MCSs (as shown by applying a precipitation tracking algorithm[36]) compared to the control simulation. However, a considerable increase over the Mediterranean in winter is noteworthy (Fig. 2b), after a decrease in autumn (Fig. 2h). Little change in SHP is projected for spring, when changes are insignificant over roughly half of the domain based on 1000 bootstraps (Fig. 2d). Annually, a negative signal is dominant over most of the continental Europe and parts of the Mediterranean, and little increase is evident in the north (Fig. 2j).

Note that there are some interesting responses over northern Africa, but we do not consider these here due to their proximity to the lateral boundary of the model.

In order to compare these findings with results corresponding to a lower degree of global warming, a simulation for 2040-2049 has been conducted using the same model configuration. This mid-century simulation is also for RCP8.5 but gives insight into changes for lower emissions as the greenhouse gas forcing in the driving GCM in the 2040 s is more moderate (as are the sea surface temperature changes introduced in the model). Overall, the SHP in the 2040 s reveals a decrease compared to the current climate, but the decrease is lower than that for the end-of-century simulation (Supp. Fig. 4). The enhanced decrease with time is evident even at the regional/seasonal level, with reductions in central and SEU in summer and autumn increasing in magnitude between the mid-century (GMST change of +2.3 K in the driving GCM, compared to pre-industrial period) and end-of-century (+5.2 K GMST increase) (Table 1). Some localities near the Mediterranean coasts (southern France, offshore eastern Spain, and parts of Central Mediterranean Sea) show increasing frequency of SHPl in autumn in the mid-century simulation, but these diminish by end of century, mainly due to further reductions in summer (Supp. Fig. 4). This is likely due to the different evolution of competing mechanisms with time. A monotonic change with time is projected for wintertime increases in SEU and summertime decreases in Britain (Table 1). Results are less consistent between the two periods for Northern Europe (Table 1); however, the quantitative values in these regions are much lower than in Central and Southern Europe (CEU and SEU, respectively).

## Changes in very large hail and beyond
We find that "SSHP", indicating the possibility for very large hail-stones, also decreases around Central Europe, despite a small increase in Graupel50 (Fig. 3). It also remains low over the British Isles and Northern Europe land areas (Fig. 3). In contrast, however, it increases in SEU in autumn and winter, balancing decreases in summer and spring (Fig. 3). Overall, significant SHP shows a much lower future decrease than SHP.

observed hail frequencies for Central Europe (of the order of 1000 cases y[-1] in Central Europe[29]).

We find that for warmer RCP8.5 end-of-century conditions, "SHP" cases reduce by more than half in summer (Fig. 2, right column), despite little change in supercell frequency, as estimated by an updraught helicity metric (Supp. Fig. 3a). The relatively stable number of supercells indicate that what matters more here is the

**Table 1 | Regional counts of Severe Hail Potential for the control (1998–2007), mid-century (2040–2049), and end-of-century (~2100) simulations**

|  | BRI control | BRI 2040 s | BRI ~ 2100 | NEU control | NEU 2040 s | NEU ~ 2100 | CEU control | CEU 2040 s | CEU ~ 2100 | SEU control | SEU 2040 s | SEU ~ 2100 |
|---|---|---|---|---|---|---|---|---|---|---|---|---|
| DJF | 0 | 0 | 4 | 0 | 0 | 7 | 0 | 0 | 4 | 178 | 614 | 1477 |
| MAM | 16 | 96 | 32 | 284 | 68 | 94 | 4003 | 2388 | 1117 | 3270 | 3386 | 3396 |
| JJA | 188 | 154 | 81 | 694 | 1075 | 732 | 6247 | 2625 | 657 | 11443 | 5554 | 2193 |
| SON | 6 | 1 | 17 | 11 | 3 | 43 | 568 | 178 | 24 | 8144 | 7390 | 4449 |
| ALL | 210 | 251 | 134 | 989 | 1146 | 876 | 10818 | 5191 | 1802 | 23035 | 16944 | 11515 |

Global mean surface temperature (GMST) change for the global circulation model driving the mid-century simulation is +2.3 K, and for the end-of-century simulation is +5.2 K, compared to the pre-industrial period. BRI refers to British Isles, NEU refers to Northern Europe, CEU refers to Central Europe, and SEU refers to Southern Europe, as shown in Fig. 3.

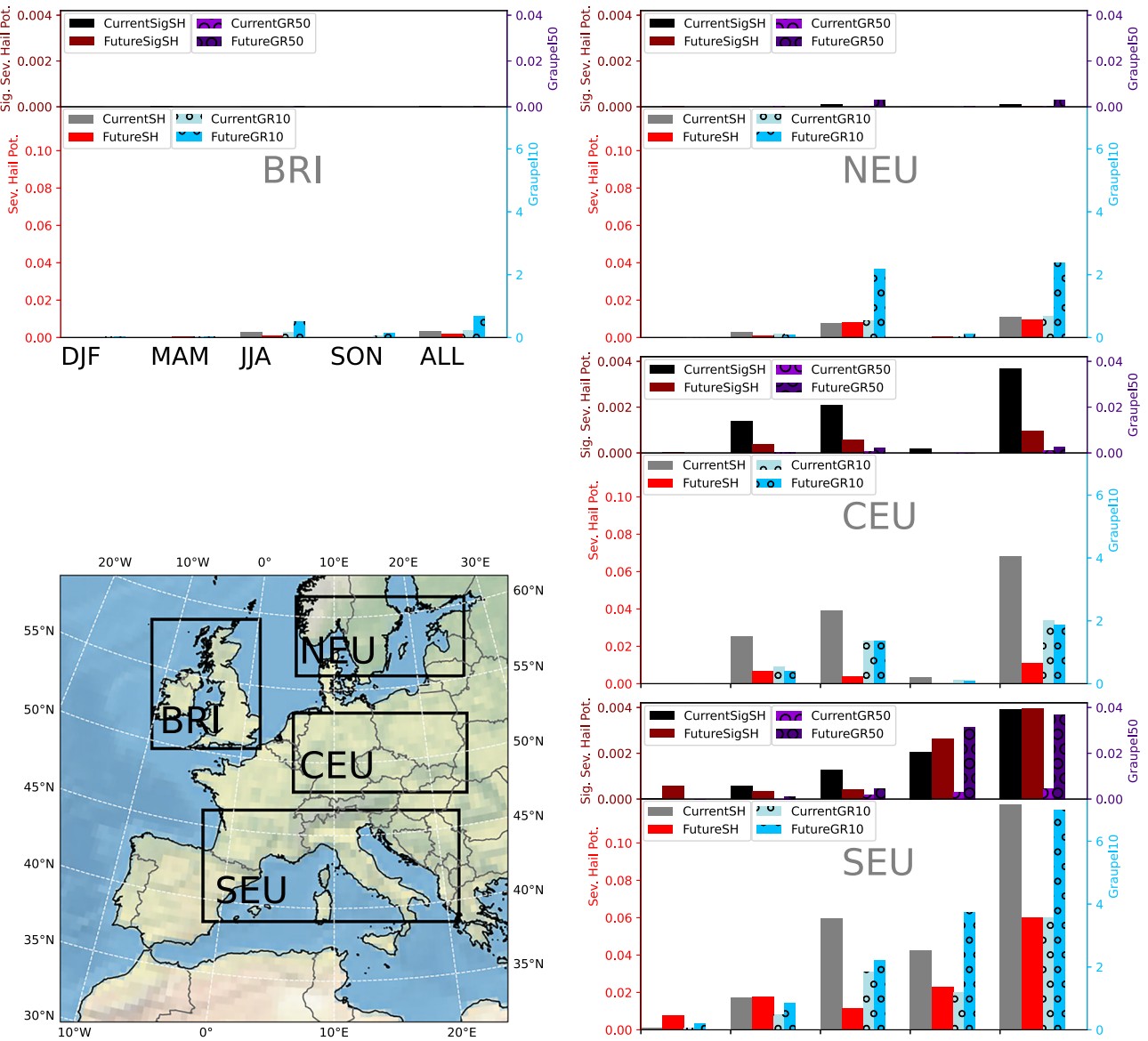

**Fig. 3 | Regional frequencies of severe hail.** Regional frequencies of severe hail potential (SH), significant severe hail potential (SigSH), and days with maximum graupel water path exceeding 10 kg m⁻² (Graupel10, GR10) and 50 kg m⁻² (Graupel50, GR50) for four seasons (December, January and February: DJF, March, April and May: MAM, June, July and August: JJA, September, October and November:

SON), and all year (ALL) in the current and future (end-of-century) simulations. Calculated data are from only land grid points, and frequency values are per km² per year. BRI refers to British Isles, NEU refers to Northern Europe, CEU refers to Central Europe, and SEU refers to Southern Europe, the associated boundaries are shown in the map as black boxes.

**Table 2 | Significant Severe Hail Potential (SigSHP) to Severe Hail Potential (SHP) ratio for control and future (end-of-century) simulations, and percentage change for different regions**

|  | Control SigSH/SH | Future SigSH/SH | Change % |
|---|---|---|---|
| BRI | 0.0095 | N/A | N/A |
| NEU | 0.0091 | 0.0011 | −87.5 |
| CEU | 0.0538 | 0.0855 | 58.9 |
| SEU | 0.0324 | 0.0657 | 102.7 |

British Isles (BRI), Northern Europe (NEU), Central Europe (CEU), and Southern Europe (SEU), as shown in Fig. 3.

Greater local increases in significant SHP stem from larger increases in vertical velocities in the HGZ, while wind shear remains considerably high in the future autumn (Supp. Fig. 5): e.g., no HGZ vertical velocities exceeding $30 \, m \, s^{-1}$ in the $10 - 30 \, m \, s^{-1}$ deep layer shear range in the current autumn climate, whilst several occurrences in the future. This is consistent with the increases in supercell density around Mediterranean coastal areas in the autumn (Supp. Fig. 3); supercells tend to occur in environments with high deep layer shear[37], and are expected to be major producers of significant severe hail[1,38,39]. In winter, storms with SHP tend to have both higher HGZ vertical velocities and higher deep layer shear in the future (Supp. Fig. 5).

The ratio of significant severe hail to SHP increases except for the north (Table 2), where the frequencies overall are very low. In SEU, this ratio doubles, and over Central Europe (CEU) it is 1.5x greater (Table 2). This suggests a shift towards conditions favourable for the formation of the largest hailstones despite a decrease in SHP in a warmer climate, especially further south in Europe. As discussed below, there is a missing process in the proxy -namely the reduced melting of larger hailstones due to higher fall speeds- which could favour further increases in larger hail. Hence, it is likely that the significant severe to SHP ratio here (Table 2) represents a low estimate of the possible shift to larger hail at the surface.

Graupel50 shows an 8-fold increase in SEU, mainly in the autumn, and to a lesser extent in summer (Fig. 3). This increase is consistent with the increase in significant SHP in autumn over the Mediterranean, but other factors for hail formation, such as those captured by the proxy, i.e. time spent in the hail growth zone, are also important.

## Emergence of warmer thunderstorms in the future

In autumn (and summer), the distribution of freezing level heights in gridpoints satisfying all other criteria of the SHP proxy becomes bimodal in the future, suggesting the emergence of a second class of thunderstorms with higher freezing level heights ( >4.5 km Fig. 4c, d). Bimodality is evident even in the annual distribution (Supp. Fig. 6).

It is possible these warm-type thunderstorms could be important for producing significant severe hail, as, despite the high freezing levels, the largest hailstones may still reach the surface due to their high terminal velocity. This effect is not included in our current proxy, but if included could provide a mechanism for an increase in the occurrence of very large hail in Europe. If we relax the criteria in freezing level height (allowing up to 5.5 km) to incorporate these warmer thunderstorms, then the overall future decrease in SHP would be 23.5%, compared to the 57.0% reduction with the original proxy.

To support our argument, there is some observational evidence of hail from warmer types of convective storms in the tropics[9,40]. Although ground observations are limited, satellite-based estimates do suggest hail in such areas[41,42], with a few cases where hail was associated with very high freezing level heights in India[43], Taiwan[44] and West Africa[45].

Based on this, we modify the SHP proxy to identify candidates for warm-type thunderstorms in our European simulations by constraining the freezing level height to lie between 4.5 and 5.5 km and removing the deep layer threshold (since it is found that hail-producing tropical thunderstorms usually have <10 m s⁻¹ deep layer shear[9], and later confirmed for maritime tropical hail[40]). In the future, we find that potential warm-type thunderstorms occur throughout the warm season, being most frequent in September (Fig. 4e). Between July and October, the number of candidate warm-type thunderstorms exceeds the number of storms identified by the standard SHP proxy (by a factor of 4 in September). The warm-type thunderstorms occur mostly in SEU, particularly in Italy in summer and autumn (Fig. 5a, c), and across most of the Mediterranean Sea and adjacent coastal areas in autumn (Fig. 5c). The impacts of hailstorms could regionally increase in Italy and its surroundings, if these warm storms are associated with hail.

Similar signals are evident if the threshold for vertical velocities in the HGZ is increased, from 10 m/s to 15 m/s (Fig. 5b, d, f). The likelihood of very large hailstones, which are less prone to melting, increases at higher vertical velocities, typically with the higher convective available potential energy associated with warmer environments.

To elaborate further, we calculate the frequency of Widespread Very High Instability (WVHI) cases in our simulations, defined as Lifted Index < -25 K over an area covering at least 10,000 km² for a given output time (Fig. 6). The present-day control simulation does not feature any such extreme conditions, and there are none in the future simulation in the cold season between November and May. However, these do exist in end-of-century summers and autumns as episodes lasting several consecutive days (for example, there are 6 episodes of WVHI which last for more than 7 days). Such episodic extremes are in line with current severe hailstorm episodes observed during or after recent heatwaves in Europe, and are mainly associated with positive SST anomalies in the Mediterranean Sea[46]. In the current climate, seas surrounding the Arabic peninsula in the Middle East feature very high SSTs and >5000 J/kg CAPE during summers, but very few thunderstorms occur in these areas due to high CIN and limited convection initiation; and the wind shear is weaker in the subtropics. In SEU however, cut-off lows, mesoscale disturbances, and steep mountains help convection initiation, and stronger shear helps organising the thunderstorms formed.

Therefore, we propose that the occurrence of warm-type thunderstorms in SEU in a future warmer climate could magnify the impact of hailstorms in Italy and surrounding areas with more frequent significant severe hail-favourable conditions, if such storms do indeed produce very large hailstones (noting there are observational precedents from thunderstorms in the tropics).

## Variability and drivers for changes

We find that SHPl in winter is much more variable in the future, compared to the current climate, while summer variability decreases (Fig. 4e). There is a notable future shift of the peak hail month from June towards May over land (Supp. Fig 9) and from October to December over sea (not shown). Decreases in warm season variability stem from a projected shift in thunderstorms towards warmer FZLV ranges.

SHP decreases in the future over most of continental Europe in summer (Fig. 2f) and parts of the Mediterranean in autumn (Fig. 2h). To identify the drivers of these changes, we first compute future changes in the distributions of each ingredient of the SHP proxy, using data from all grid points for freezing level height and deep layer shear, and only points with graupel water path >100 g m⁻² for vertical velocity in the HGZ (to pick up locations/times of sufficient updraught speed when a hail embryo -graupel- exists). The results are shown by the orange lines in Fig. 7, from which it follows that freezing level heights increase in future summer and autumn, while deep layer shear and vertical velocities in the HGZ decrease.

For each ingredient, we then compute future changes in the distribution constructed using data from candidate storm points only (i.e.

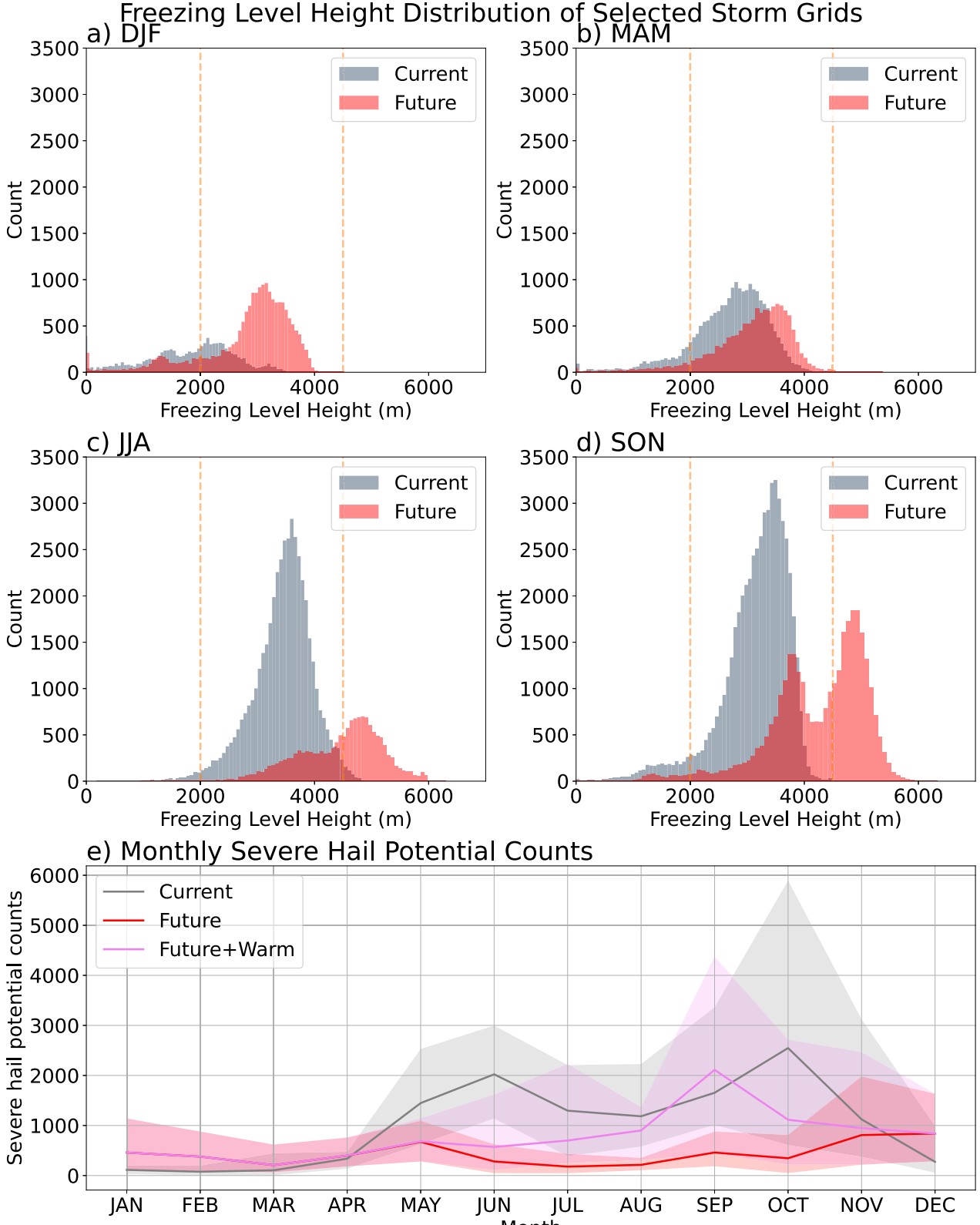

the subset of grid points remaining once the relevant constraints on the other proxy ingredients have been imposed), shown by the purple lines in Fig. 7. It is evident that future changes in the distributions of SHP ingredients at potential storm locations are qualitatively similar to changes in the underlying full distributions. This is true for all ingredients, suggesting that large-scale future increases in freezing level height and future decreases in deep layer shear and vertical velocity in

the HGZ in summer and autumn all contribute to the future decreases seen in SHP. Note that, in other seasons, future changes in the distributions of proxy ingredients at candidate storm gridpoints do not always follow changes in the full distribution. For example, in winter, the distribution of deep layer shear in storms shifts towards lower values in the future, whereas the full distribution shifts towards higher values (Supp. Fig. 7).

**Fig. 4 | Freezing level height analysis and monthly distribution.** Freezing level (FZLV) height distribution of grid points satisfying Severe Hail Potential criteria, except the FZLV height component. These all simultaneously exceed the graupel, vertical velocity in the hail growth zone, and vertical wind shear thresholds (as defined in severe hail potential); for **a** December, January and February (DJF), **b** March, April and May (MAM), **c** June, July and August (JJA), **d** September, October and November (SON). Storms with FZLV height between 2000 and 4500 m (those between the dashed orange lines) satisfy the Severe Hail Potential criteria.

"Current" refers to 1998–2007, and "future" refers to the end-of-century. **e** Severe hail potential per month in the current and future (end-of-century) simulations. The grey and red shaded areas span the 10 years sampled within the current and future (end-of-century) simulations, respectively, and lines in (**e**) depict the average values. Similarly, violet shading and line show the total of future severe hail potential if the warm-type thunderstorms were included. The analysis is for the whole domain analysed (i.e,. excluding 70 grids from each lateral boundary).

In general, it is not possible to quantify the relative importance of future changes in each proxy ingredient to the overall changes in SHP, as the set of storm grids selected will change while relaxing any of the proxy criteria. However, in the important case of SHP decreases in summer and autumn, some progress can be made by noting that, in the present-day, the freezing level lies between 2 and 4.5 km at nearly all (97%) gridpoints, satisfying the other criteria of the SHP proxy (Fig. 4c, d). Therefore, if the freezing level height constraint is removed from the proxy (which we define as Experiment 1, or EXP1 for short), essentially the same set of storms will be identified in the present-day, implying that future changes in EXP1 can be meaningfully compared with those in the original SHP proxy, in these seasons, i.e., we can explore to what extent the increase in freezing level height is a major driver of the overall decrease in hail.

Future changes in the modified EXP1 proxy are shown in Supp. Fig. 8. In summer and autumn, future decreases are also apparent in EXP1, with a similar spatial pattern to the decreases in the full SHP proxy (Fig. 2). This implies that thunderstorm candidates with high vertical velocities in the HGZ and/or high deep-layer shear are less frequent in the future. Supp. Fig. 8 shows absolute relative differences between future changes in EXP1 and SHP in summer and autumn; regions where these differences exceed 50% correspond to those where future decreases in SHP are dominated by increases in freezing level height. From Supp. Fig. 8 it is apparent that differences between future changes in EXP1 and SHP are typically small in regions exhibiting the most pronounced future decreases in SHP, implying that future changes in SHP are driven mainly by decreases in vertical velocities in the HGZ and/or deep layer shear. Yet, there are some interesting local exceptions. For example, over Italy, future decreases in SHP in summer are dominated by increases in freezing level height and thus increased melting. Prior research[27] also find increasing melting level heights across the CONUS but note that the decrease in ≥2 cm hail does not stem from this feature. We note also that the graupel-based proxy does not include the effect of increased melting.

Overall, we conclude that less frequent SHP in the projected future climate is not due to a single factor. Weaker updraughts in the HGZ, increased melting associated with higher freezing levels, and weaker deep-layer shear can all contribute to the reduction, and the relative importance of different drivers varies spatially. Weaker updraughts in the HGZ in the future may stem from increases in the HGZ height, and possibly from reduced updraught strength due to dry air entrainment with reduced mid-tropospheric relative humidity in future storm environments[35], or from additional precipitation loading[47]. Reduced deep layer shear may also lead to narrower updraughts and more entrainment. We note that decreases in SHP occur despite an overall increase in instability with warming[30]. This rise in instability alone would favour stronger updraughts, which may skew the hail size distribution towards larger (very large) hailstones when other conditions are met.

## Discussion

We have used a pan-European 2.2 km convection-permitting climate model to examine future changes in a physically-motivated, ingredients-based proxy for hail, and have found that the potential for severe hail generally decreases in the future. This contrasts with previous studies, which project robust increases for both severe and significant severe hailstorms in the future across Europe[25], increases in damaging hail-relevant weather types around Germany[48], and slight increases in the potential for hail across Germany[49]. For the UK, we find little future change in the frequency of SHP, while a decrease was suggested in an earlier study[50]. Differences, at least partly, stem from the study methodologies. In particular, previous studies do not include the effects of warming on HGZ level changes, freezing level height changes, entrainment, precipitation loading, etc., that counteract the effects of increases in instability; previous studies solely focus on environmental conditions rather than simulated convection, thus increases in instability with atmospheric warming dominate their results. In our case, the increase in instability is still there[30], hence our model is not producing a conflicting signal for the environmental conditions, but the different results come from the hidden detail in the simulated storms (e.g., less frequent strong updraughts in the HGZ, which is higher in warmer profiles, and weaker shear). The use of a standard instability metric (e.g., MUCAPE) might not capture details in the storm profiles well, and what matters more is what is happening in the layer where the hail growth occurs.

Although SHP is projected to decrease, we have found evidence for the emergence of a category of warm-type thunderstorm in the future. These storms may produce hailstones large enough to reach the surface, despite increases in freezing level height. We expect a future shift in the hail size distribution towards larger hailstones since we have found the ratio of significant severe to SHP increases (as also in the literature[22,51]. This is important because the impact of hailstorms increases non-linearly with hail size[13,52], and even a small increase in size could reverse the effects of a decreasing signal in hailstorm frequency. However, there is considerable uncertainty regarding the effect of enhanced melting associated with higher freezing levels on the largest hailstones. Further studies of these warm thunderstorm clusters are therefore needed to improve our understanding of their potential to produce very large and damaging hail at the surface, perhaps by combining observational field studies and dedicated very high-resolution idealised simulations.

Our results are consistent with the U.S. study performed with convection-permitting simulations[27], in terms of regional decreases in hail in summer, a seasonal extension (longer warm season, with hail-storms extending into spring and autumn), and changing signals with hail size (favouring more large hailstones, also seen locally[22]). Another recent U.S. study found that differences in warm cloud depth are important for future changes[53]; this is consistent with our freezing level height results.

It is important to note that CPMs do provide more insight into future changes in convective storms, as they allow a shift from an environments-based approach towards a simulated storm-scale processes paradigm[30]. Indeed, simulated updraughts and storm-scale features are found to be more skilful predictors compared to environmental fields for severe convective weather[54], and these differences have the potential to lead to dramatic changes in the future projections of individual hazards.

Convection-permitting climate simulations are currently feasible for regional to local studies only. Downscaling multiple GCMs can help to quantify uncertainty in future hail changes due to uncertainty in the large-scale atmospheric conditions. This is arguably the most important aspect when analysing severe convective

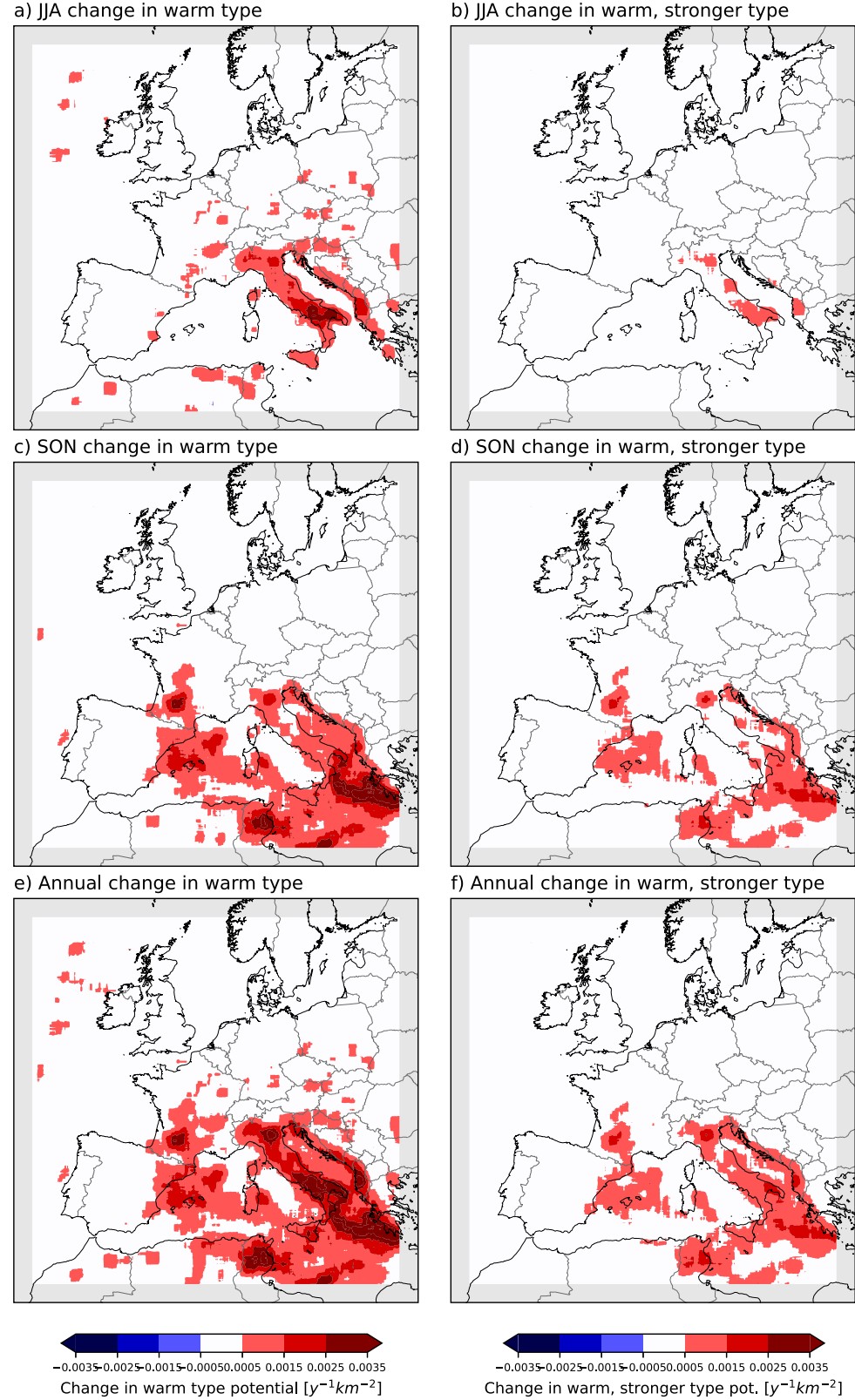

**Fig. 5 | Warm-type thunderstorms.** Future (end-of-century) change to the "warm type" thunderstorms for June, July and August (JJA, **a** September, October, and November (SON, **c** and whole year (**e**). Warm type thunderstorms are defined using the same severe hail potential proxy but with freezing level height values between 4500 and 5500 m, and with no shear criteria. **b**, **d**, and **f** are the same but for changes in "warm type" thunderstorms, which have stronger updraughts in the hail growth zone (using 15 m s⁻¹ threshold instead of 10 m s⁻¹).

storm hazards within a changing climate, as their characteristics have been shown to be extremely sensitive to circulation changes and other larger scale factors[36]. A limitation of our study is that it is only based on only one possible realisation of the future climate, so further analysis with different climate models is needed to establish if these results are robust. In addition, we have only considered a single emissions scenario (RCP8.5). However, by comparing mid-century (global warming level of 2.3 K) and end-of-century (global warming level of 5.2 K) simulations, we have found that changes in hail potential approximately scale with global warming, suggesting that

the results presented here could be translated to lower emissions scenarios with appropriate scaling.

Future studies with more focus on storm-scale processes and individual storm features, such as storm type, updraught width, storm-relative flow, and a coupled storm-tracking solution might help in terms of capturing changes throughout the storm lifetime rather than instantaneous values at discrete times[7], perhaps using models with even higher (<1 km grid spacing) resolution[55].

## Methods

### Climate simulations

The Unified Model (v10.1) climate simulations were run at the UK Met Office Hadley Centre, to assess future changes across Europe with convection-permitting resolution[56], driven by the N512 HadGEM3 global simulations[57]. These include 10 years for the current (1998–2007), and 10 years for the mid-century (2040-2049, under RCP8.5) and end-of-century future (~2100, under RCP8.5), excluding a 1-year spin-up period for each. The model configuration features ~2.2 km horizontal grid intervals, with a rotated-pole grid structure to optimise distances across Europe. There are $1536 \times 1536$ horizontal grid points, 70 vertical levels in the atmosphere, and 4 soil levels in the model domain. To remove lateral boundary artefacts, an external rim of 70 grid points is excluded from this analysis.

The model used has the "ENDGame" dynamical core[58]. For physical parameterisation settings, Met Office operational UKV model physics[59] with updated microphysics and planetary boundary layer schemes[60,61] are used. No cumulus parameterisation is set, as all convection is assumed to be explicitly resolved with a 2.2 km grid. Sea surface temperature for the future simulation is derived from 1999–2008 observations, by adding 20-year mean "delta changes" which are extracted from global coupled model simulations for 1990–2010, 2040–2049, and 2090–2110. Further information on the

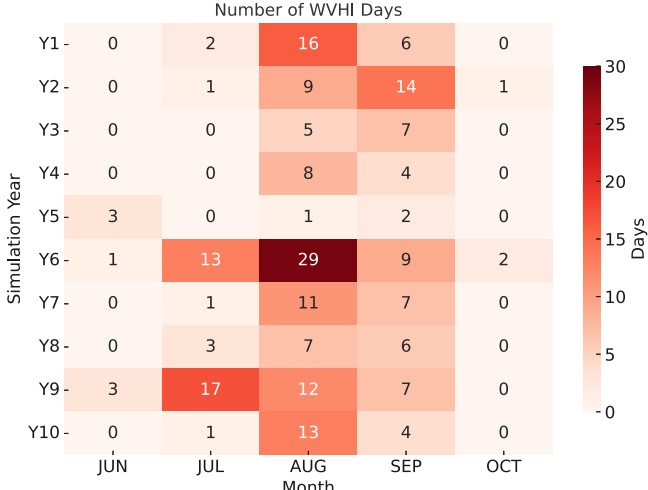

**Fig. 6 | Widespread Very High Instability frequency.** Distribution of Widespread Very High Instability (WVHI) Days in the future (end-of-century) simulation.

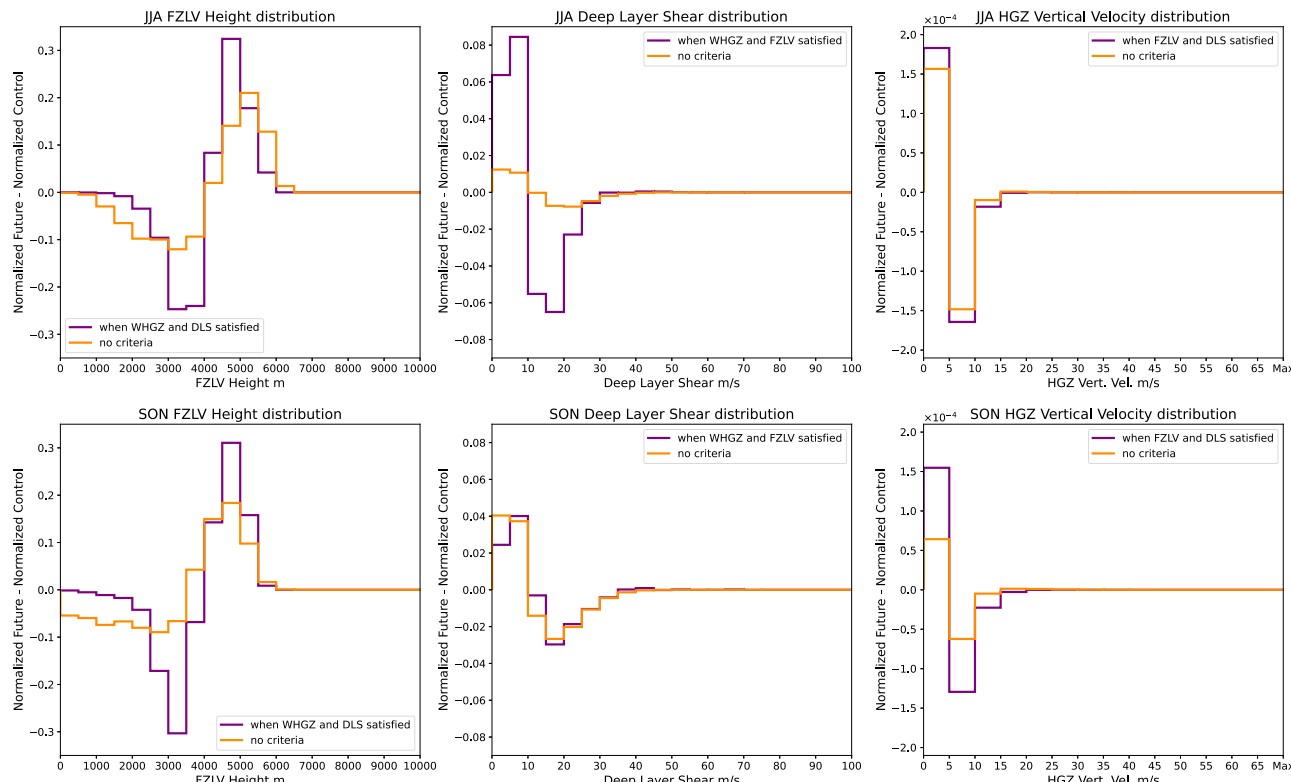

**Fig. 7 | Change in proxy components.** Future changes in normalised distributions of Severe Hail Potential proxy components [0-degree level height, Deep Layer Shear, and HGZ (Hail Growth Zone) Vertical Velocity with graupel, from left to right], when applied separately, and when restricted by the criteria of the other components of the proxy [June, July and August (JJA, top) and September, October and November (SON, bottom)].

model configuration and details on its improvement in precipitation extremes compared to coarser resolution can be found in the literature[56,62].

## Analysis of model data for assessing Severe Hail Potential

The SHP proxy is a newly developed approach, which is evaluated in an earlier study[29] in detail, using a 20-year long hindcast simulation, which possesses an identical configuration to the three climate simulations used in this study (but using ERA-Interim reanalysis as input, instead of the driving GCM). Here, we briefly describe the proxy and its basis. Instantaneous data on pressure levels at 3-hourly intervals is used to calculate hail-focused parameters. An explicit hail parameter doesn't exist in the stored model output, but graupel does. The vertical velocity within the hail growth zone is determined as the maximum vertical velocity value between −10 °C and −30 °C of the atmospheric column for each grid. This is calculated for each grid cell with at least 100 g m$^{-2}$ graupel water path, and a threshold of 10 m s$^{-1}$ used to indicate conditions favourable for larger hail growth.

Deep layer shear is well-known to be crucial in severe hail producing storm environments[5,63,64], not only because of its role in storm organisation, but also due to the fact that higher wind shear values lead to wider updraughts (hence more favourable conditions for larger hail), which is supposed to be an outcome of stronger low-level storm-relative winds[6,65–67]. Deep layer shear is defined as the absolute value of the vectoral difference of the low-level horizontal wind vector and the 500 hPa horizontal wind vector. The choice of 500 hPa instead of e.g., 6 km above ground should affect the results over higher terrain, by having less shear, but the higher the mountains, the less deep the storm clouds are, which would tend to have the reverse effect. Thus, we believe the 500 hPa choice here is optimal. Here, the low-level vector is determined as the lowest available pressure level depending on surface elevation, either 925, 850, or 700 hPa. We do not use surface winds explicitly as these may be locally modified due to downdrafts of convective storms simulated by the model, which would potentially misrepresent the kinematic environment of the storm (we aim to look for the environmental shear, rather than the local shear produced by the storm itself). New research suggests that straighter hodographs (meteorological diagrams which depict vertical profiles of horizontal winds through an atmospheric column for a given time) lead to more prolific hail production through wider updraughts[7], and the speed component in high wind shear environments are more favourable for large hail, rather than the directional shear component. The lowest layer tends to have more directional shear than upper levels. Thus, taking the low-level wind vectors slightly away from the surface might be more representative for hail-favouring conditions, compared to using 10 m winds. In addition to this, another study finds that strong low-level wind shear could be unsupportive for severe hail, while deep-layer shear starting from 1 km above surface upwards is more important[68]. Hence, we believe our approach regarding vertical wind shear is an optimum choice consistent with the latest theory favourable for large hail generation, and the availability of 3-dimensional data from our model output.

Having environments with 10 m s$^{-1}$ or larger deep-layer shear (based on the literature[5,63,69,70], combined with freezing level height between 2000 and 4500 m[23,71,72], the instantaneous updraughts satisfying the hail growth zone vertical velocity and graupel thresholds are considered as "SHP". This proxy was first applied to a 20-year hindcast simulation with the same model settings but driven by ERA-Interim data[29]. This was found to agree well with severe hail observations from the European Severe Weather Database (ESWD[12]) in terms of the seasonal and diurnal distribution on land, as well as the spatial distribution locally (at least in regions where the observations are expected to be more representative of the ground truth).

Severe hail is a rare hazard for a locality, so an analysis in a 2 km grid, with 10-year worth of data is very noisy. Hence, we apply a spatial smoothing technique, visualising frequencies per 10,000 km2 (i.e., per 100 × 100 km). This is done by averaging the neighbouring ± 25 grid points in each direction.

SHP corresponding to the multiple of the hail growth zone vertical velocity and deep layer shear values equivalent to or higher than 400 m² s$^{-2}$ is defined as "Significant Severe Hail Potential". The result is a subset of cases from the first proxy, consisting of the top 3% cases in terms of severity. In the past, the product of most unstable cape (CAPE) and 0–6 km bulk shear (SHEAR) or WMAX x SHEAR (WMAX, theoretically maximum vertical velocity being derived by CAPE empirically) is used for such purposes with coarse-resolution model output. Here, the vertical velocity directly resolved by the convection-permitting simulations can be considered as a much more representative value in hail formation (given that they only occur on the storm grids, rather than representing a wide environment, and are dynamically resolved with many additional factors involved in their numerical calculations), compared to CAPE-based estimates, such as WMAX from coarse-resolution models; hence, this is used here as an improved way to assess favourable storm formation (with higher severe hail likelihood). The top 3% of cases are chosen because the observed frequency of very large hail ( ≥5 cm diameter) with respect to large hail ( ≥2 cm diameter) is around this value[13,29]. Earlier studies suggested values, such as ~6–8%[13,73,74], but it is acknowledged that the likelihood of reporting much larger hailstones compared to smaller ones (albeit still in the large category) is higher, especially for historical cases, as almost all observational studies confirm[74,75].

Based on the climatology built using the CPM, severe hail occurs more frequently in SEU, and the cases gradually decrease towards the north[29]. Sea areas (excluding the Mediterranean) experience very little to no hail[29]. The Mediterranean Sea on the other hand, has higher frequency, especially around the Ionian Sea[29]. Land areas have peaks from May to August, and the Mediterranean in the autumn[29]. The SHP proxy produces a reasonable seasonal and diurnal distribution of severe hail in Europe (Supp. Fig 1), with peaks in summer afternoons (between 12-21 UTC). In autumn and winter as expected, there is less of a diurnal cycle in hail, with this occurring almost exclusively over the sea. A comparison with the frequency of ground observations around Central Europe, where the European Severe Weather Database (ESWD) is more representative, suggests a reasonable match (Supp. Figure 2). Although the SHP is higher than the observations in all seasons, it is still of the same order of magnitude in summer, although differing more in the spring and autumn. This may be due to less underreporting or more hail vulnerability in the summer, if not from model and proxy biases. ESWD data is extracted to remove unchecked reports (QC0), and only the reports with maximum hail size ≥2 cm are used.

## Significance testing

Significance testing for changes in SHP is performed via non-parametric bootstrapping. 1000 random resamples of the smoothed data, for each of current and end-of-century future seasons, are produced (resampling as seasonal blocks). Future changes are calculated for each of the 1000 samples, with their corresponding counterpart. Then, the changes are sorted and, if the 5−95th percentile confidence interval overlaps zero, the change is deemed not significant at the 10% level. Grid points with insignificant signals based on this approach are masked in white.

## Supercell proxy

Updraught helicity is defined as follows:

$$UH = \int_{z0\,=\,2km}^{z1\,=\,5km} w\zeta\,dz \qquad (1)$$

where w is the vertical component of velocity, ζ is the vertical component of vorticity, z is the depth of layer(s), and the vertical integration over the 2–5 km layer above ground level[76].

For the supercell proxy here, we modify this such that instead of the 2–5 km layer, a layer between 850 and 500 hPa is used:

$$UH_{modified} = \int_{z0 = z_{850hPa}}^{z1 = z_{500hPa}} w\zeta \, dz1$$

This is so it can be used with available model data.

## Data availability
The data generated in this study have been deposited in a database at https://doi.org/10.5281/zenodo.16325483. The Met Office model data used are under Crown copyright of the UK government, and access may be requested from the Met Office. Due to funder restrictions, the raw model data is currently not publicly available.

## Code availability
Analysis code used in this paper is available from the corresponding author upon request.

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

## Acknowledgements

This work is supported by the UK NERC-funded FUTURE-STORMS Project (NE/R01079X/1, HJF and AK). Elizabeth J Kendon gratefully acknowledges funding from the European Union under Horizon 2020 project European Climate Prediction System (EUCP; Grant Agreement: 776613) and from DSIT funded Met Office Hadley Centre Climate Programme (GA01101). Chris J Short is funded by GA01101. Segolene Berthou is acknowledged for the mid-century simulations, and Colin Manning for his comments on the manuscript.

## Author contributions

A.K. performed the analysis, generated the visualisations, and drafted the original manuscript. E.J.K. carried out the simulations. H.J.F. secured funding for the project. H.J.F., E.J.K., and C.J.S. advised on the analysis and visualisations, and co-wrote the paper. All authors were involved in developing the methodology, have participated in discussions, and

contributed to editing successive draughts, approving the final version of the manuscript.

## Competing interests

The authors declare no competing interests.
