## [Transparent Peer Review file · Nature Communications]

Future changes in severe hail across Europe including regional emergence of warm-type thunderstorms

Corresponding Author: Dr Abdullah Kahraman

This manuscript has been previously reviewed at another journal. This document only contains information relating to versions considered at Nature Communications. Mentions of the other journal have been redacted.

Version 1:

Reviewer comments:

Reviewer #1

(Remarks to the Author)

I thank the authors for their responses to my previous review for this article submitted to [REDACTED] The reviewer's responses have markedly improved the manuscript. Please refer to attached pdf review for further comments.

[Editorial Note: Please see the attachment at the end of the file]

Version 2:

Reviewer comments:

Reviewer #1

(Remarks to the Author)

Please see attached review.

[Editorial Note: Please see the attachment at the end of the file]

REVIEWER COMMENTS

Reviewer #1 (Remarks to the Author):

I thank the authors for their responses to my previous review for this article submitted to [REDACTED] The reviewer's responses have markedly improved the manuscript. Please refer to attached pdf review for further comments.

Review of NCOMMS-24-10209A

This review is for manuscript NCOMMS-24-10209A, titled "Future changes in large hail across Europe dominated by new warm thunderstorm type".

The study shows an evaluation of proxy-derived changes in frequency of severe hail conditions across Europe. The proxy uses features derived from a convection-permitting model rather than broader environmental analysis, yet it remains a proxy with the well-known disconnect between hail-prone conditions and hail occurrence. The results show decreases in severe hail potential across Europe, particularly in the warm season, with some cool season increases. The paper identifies that more tropical-like storms with high melting levels occur in the future scenario, but the claim that they dominate changes in large hail is untested.

The paper is an application of an innovative proxy across a large domain and the results will help to untangle the complex effects of climate change on hailstorms. It is excellent that Kahraman et al. (2024) has been published, providing justification for and detailed analysis of the proxy used in this study.

While the results of the study are useful and interesting there remain serious shortcomings with the paper that mean that, unfortunately, it requires further revision before being acceptable for publication. I hope that my suggested changes below allow for the manuscript to be improved.

The biggest issues are as follows:

1. Analysis of change drivers: In a previous review of this manuscript I wrote that the authors' method for analysing the drivers of changes was flawed. The authors' responded that running new simulations is too computationally expensive, which I acknowledge is a difficulty (other approaches such as removal of biases in ingredients could be considered). The problem of unfounded claims regarding the drivers of changes remains in the revised manuscript.

In particular, on lines 260-275, the results (EXP1 and EXP2) are interesting but they still do not show what the authors claim in the text. As the authors' acknowledged in their responses, as soon as the proxy is changed by removing threshold(s), the proxy is no longer detecting the same storms but rather a larger set of storms, including those that were not detected because of the thresholds in the original proxy. While changes in this larger set may

imply driving factors, as the authors wrote in their responses, any changes do not necessarily explain changes in the original set. As I wrote in my previous review, to properly determine the driving factors for the observed changes, the authors would need to run the same proxy on simulations in which changes to certain aspects (e.g. FLZ) had been removed. Alternatively, the authors could use their existing data and look at changes in individual proxy ingredients for the set of storms identified using the original proxy to see which ingredients had the largest changes (this too is imperfect since the largest changes may not be the most explanatory).

There is useful information to be found by comparing changes using different proxies, as the authors have done, but care must be taken in how those changes are described. This is because the modified proxies may no longer detect only severe hail conditions and therefore cannot be used to show the drivers for changes in severe hail. For example the sentence on line 268 "We find that a decrease in the occurrence of high vertical velocities in the HGZ is the dominant factor leading to a decrease in severe hail" is not backed up by the evidence shown in the paper. On the other hand, an earlier sentence starting on line 262 "On removal of the freezing level height parameter from the proxy, future decreases are still seen for all seasons (Supp. Fig. 8), which implies that future thunderstorms with high shear and high vertical velocities in the HGZ are less frequent" is fine.

We thank the reviewer for their constructive review, and apologize for the delay in the response. We reassess our methodology and analysis in response to changes in drivers. As the reviewer notes, conducting a perfect analysis is challenging. Our experiments were encompassing a broader range of storms than the original Severe Hail Potential proxy, which, while useful, remained an imperfect tool for understanding these drivers.

Regarding their first suggestion, we are not clear what the reviewer was alluding to. Although they acknowledge running our simulations are too expensive in their comment, it sounds like they are suggesting additional simulations (perhaps PGW-like experiments), but unfortunately we don't have the capacity to do this as our model is very expensive.

The second suggestion of the reviewer is to use our existing data and look at changes in individual proxy ingredients for the set of storms identified using the original proxy to see which ingredients had the largest changes. We interpret this as analysing the change in internal distributions of individual proxy components in the control and future simulations, within the proxy thresholds.

As a more comprehensive analysis, we opt for performing this for individual proxy components, and further for all other possible combinations of these components with each other. A new figure (now Figure 7) is added to the original manuscript, showing future changes in normalised distributions of Severe Hail Potential proxy components. These are FZLV height, Deep Layer Shear, and HGZ Vertical Velocity with graupel, from left to right), applied separately, shown in orange colours. We then compute future changes in the distribution constructed using data from candidate storm points only (i.e. the subset of grid points remaining once the relevant constraints on the other proxy ingredients have been imposed), shown by the purple lines. Only JJA (top) and SON (bottom) are shown here for clarity (we put a more comprehensive version having all seasons and whole year in the supplement, discussed later).

The freezing level heights increase in future summer and autumn regardless of the proxy criteria are satisfied or not. This is also valid for deep layer shear and vertical velocities in the HGZ, they all decrease during the storms and other times; suggesting that large-scale future increases in freezing level height and future decreases in deep layer shear and vertical velocity in the HGZ in summer and autumn all contribute to the future decreases seen in SHP.

A more comprehensive figure is added to the supplement as Supp. Fig. 7, which shows the same for all seasons, but with all combinations of the proxy components included as well. This figure allows analysing relative distributions of individual proxy components, when one or more other criteria are satisfied. Note that, in other seasons, future changes in the distributions of proxy ingredients at candidate storm grid-points do not always follow changes in the full distribution. For example, in winter, the distribution of deep layer shear in storms shifts towards higher values in the future, whereas the full distribution shifts towards lower values (Supp. Fig. 7). This means that whilst the overall shear is weakening in future winters, it increases during the storms (which is contributing to the increase in SHP). These are mainly the Mediterranean storms.

In general, it is not possible to quantify the relative importance of future changes in each proxy ingredient to the overall changes in SHP, as the set of storm grids selected will change while relaxing any of the proxy criteria. On the other hand, we can explore to what extent the increase in freezing level height is a major driver of the overall decrease in hail, by comparing the EXP1 and Severe Hail Proxy changes in a new figure below (added as Supp. Fig. 8). In summer and autumn, future decreases are apparent in EXP1 as they are in the original proxy (as in Fig 2 in the paper), with a similar spatial pattern. This implies that thunderstorm candidates with high vertical velocities in the HGZ and/or high deep layer shear are less frequent in the future. The new Supp. Fig. 8 (below) shows absolute relative

differences between future changes in EXP1 and SHP in summer and autumn; regions where these differences exceed 50% correspond to those where future decreases in SHP are dominated by increases in freezing level height. It is apparent that differences between future changes in EXP1 and SHP are typically small in regions exhibiting the most pronounced future decreases in SHP, implying that future changes in SHP are driven mainly by decreases in vertical velocities in the HGZ and/or deep layer shear. Yet, there are some interesting local exceptions. For example, over Italy, future decreases in SHP in summer are dominated by increases in freezing level height and thus increased melting. Trapp et al. (2019) also find increasing melting level heights across the CONUS but note that the decrease in ≥ 2 cm hail does not stem from this feature.

Having this significant additional analysis, we modify the section “Variability and drivers for changes” to highlight that less frequent severe hail in the future is not due to a single factor, and the drivers may vary regionally. The previous analysis was mainly around “experiments”, such as EXP2 and EXP3. The corresponding figures in the supplementary file (EXP2, EXP3) are also removed. Overall, we now conclude that weaker updrafts in the HGZ, increased melting associated with higher freezing levels, and weaker deep layer shear can all contribute to the reduction, and the relative importance of different drivers varies spatially. Weaker updrafts in the HGZ in the future may stem from increases in the HGZ height, and

possibly from reduced updraft strength due to dry air entrainment with reduced mid-tropospheric relative humidity in future storm environments (Kahraman et al. 2021), or from additional precipitation loading (as identified by Trapp and Hoogewind 2016). Reduced deep layer shear may also lead to narrower updrafts and more entrainment. We note that decreases in SHP occur despite an overall increase in instability with warming (Kahraman et al. 2022). This rise in instability alone would favour stronger updrafts, which may skew the hail size distribution towards larger (very large) hailstones when other conditions are met.

2. The new thunderstorm type: I wrote previously that more detail is required about the new “warm thunderstorm” type that the authors describe in this work. To their credit the authors have added significant detail with references to the literature, and more good analyses.

Figure

5 does a good job of showing the increases in these “warm type” thunderstorms. What remains missing is justification for the title of the paper which says that changes in hail are dominated by this new type. To justify that headline claim, the authors need to show at least the proportion of the changes that are related to the warm thunderstorm type, and that they outweigh other changes.

The main basis for the new type is the bimodal feature in freezing level heights for thunderstorms with high graupel content in autumn – so it is not necessarily for hail-bearing thunderstorms and it is for only one season (the summer results appear marginal). It is unclear

to me why the authors have not shown the connection by testing whether this bimodal distribution also appears for times that the proxy indicates are prone to severe or significant severe hail.

We now modified the manuscript title, using “moderated” instead of “dominated”. The title is changed to: *“Future changes in severe hail across Europe moderated by new warm thunderstorm type”*. This is to represent what is in the paper better. “large” is replaced by “severe”, in terms of better consistency along the manuscript. We have also added the following as a quantitative measure showing that the frequency of new warmer storms in the future simulation is in the same order of magnitude of the reduction in severe hail potential, albeit not completely equal:

“If we relax the criteria in freezing level height (allowing up to 5.5 km) to incorporate these new warmer thunderstorms, then the overall future decrease in severe hail potential would be 23.5%, compared to the 57.0% reduction with the original proxy.”

We have also renewed Fig 4 (as below), adding the month-to-month counts of future Severe Hail Potential plus the “Warm Type” to the bottom subplot (e). This is to show how the storms would compare if the new warm type was included. The section is re-written, including:

“In the future, we find that potential warm-type thunderstorms occur throughout the warm season, being most frequent in September (Fig 4e). Between July and October, the number of candidate warm-type thunderstorms exceeds the number of storms identified by the standard SHP proxy (by a factor of 4 in September).”

We further add the notion of hailstorm impacts could increase regionally (in Italy and surroundings), based on these findings (and Fig 5):

“The warm-type thunderstorms occur mostly in Southern Europe, particularly in Italy in summer and autumn (Fig 5a and 5c), and across most of the Mediterranean Sea and

adjacent coastal areas in autumn (Fig 5c). The impacts of hailstorms could regionally increase in Italy and its surroundings, if these warm storms are associated with hail.”

The bimodal distribution shows the storms with strong updrafts in the hail growth zone, high shear, and sufficient graupel content, just as used in the hail proxy. So all the storms in the 4500-5500 m FZLV range are supposed to have a severe hail potential (in terms of HGZ vertical velocity, etc.). We hope this clarifies the last point, as we think we have already done what the reviewer requests us to do. We have put dashed orange lines to $x=2000\text{m}$ and $x=4500\text{ m}$ on the subplots, and modified the figure caption to avoid misunderstandings:

“Figure 4: Freezing level (FZLV) height distribution of grid points satisfying Severe Hail Potential criteria, except the FZLV height component. These all simultaneously exceed the graupel, vertical velocity in the hail growth zone, and vertical wind shear thresholds (as defined in severe hail potential); for a) DJF, b) MAM, c) JJA, d) SON. Storms with FZLV height between 2000 m and 4500 m (those between the dashed orange lines) satisfy the Severe Hail Potential criteria. “Current” refers to 1998-2007, and “future” refers to the end-of-century. e) Severe hail potential per month in the current and future (end-of-century) simulations. The grey and red shaded areas span the 10 years sampled within the current and future (end-of-century) simulations respectively, and lines in e) depict the average values. Similarly, violet shading and line show the total of future severe hail potential if the warm type thunderstorms were included. The analysis is for the whole domain analysed (i.e. excluding 70 grids from each lateral boundary).”

Last, but not least, the rationale for the argument of new, warm thunderstorms is better justified with emerging literature. For instance, a new study (Sari and Lasher-Trapp 2025) investigated environmental features of maritime tropical hailstorms, and found that they usually have $<10\text{ m s}^{-1}$ deep layer shear, and similar freezing level height values to our warm storm definition. This complements well with the Zou et al. 2021 study, and provides further observational evidence to our argument. We have incorporated these new findings in the revision as well.

3. Proxy vs. actual hail: Care needs to be taken across the whole manuscript that the results are not misstated as showing changes in actual hail occurrence. Rather they are showing changes in proxy-derived hail potential. As an example, the abstract starts well but then mentions the “decreases in severe hail”, when no such decreases are shown and what is shown is decreases in severe hail potential. This disconnect between hail-prone conditions and hail occurrence exists for all proxy studies, and as I noted previously, most hail-prone environments do not result in hail production (e.g. Taszarek et al., 2020, for Europe).

We have examined the hail-mentioned parts thoroughly throughout the paper, and modified the occasions which could mislead the meaning accordingly (explicitly stating the “potential” wording when needed). These can be found at the track-changes version.

4. The mid-century scenario: In the text the authors consistently refer to a date range of 1940-1949 as a “future climate”, which is very confusing. Even in the methods section the date range is given as 1940-1949. Supp. Fig 4 subplot titles indicate that the period in question is actually 2040-2049 which would make more sense. I have read the paper assuming that when the authors wrote 1940-1949 they actually meant 2040-2049. Although this is essentially a consistent typo, it added to confusion reading the paper and obviously needs clarification

and fixing.

Thanks for spotting these typos. All 5 occasions (including the methods) and the Table 1 has now been fixed to 2040-2049 period.

Specific comments

1. Line 32: The results “suggest we should be prepared for (infrequent but) much larger hail in a future warmer world.” Yet the paper’s analysis of hail size is restricted to changes in the ratio of sig. severe to severe hail environments. There is no analysis of actual hail size in the results. My opinion is that this line is overselling the results.

This sentence has been modified to avoid confusion:

“We conclude that society may need to be prepared for (infrequent but) more impactful hail in a future warmer world.”

2. Line 42: I maintain that a reference on the challenges of short-term hail forecasting is required here. Allen 2020 does deal with forecasting while Raupach 2021 does not. The sentence should read “there remain uncertainties about how it forms (Allen et al. 2020), challenges around its short-range forecasting (reference) and its projection with a changing climate (Raupach et al. 2021)”.

Now a couple of references are added to the text:

“there remain uncertainties about how it forms (Allen et al. 2020), challenges around its short-range forecasting (Kim et al. 2023, Adams-Selin and Ziegler 2016) and its projection with a changing climate (Raupach et al. 2021).”

3. Line 63: As discussed in my previous review hailstone melting can be delayed to well below the zero degree dry-bulb level (e.g. Lamb and Verlinde, 2011) and the authors should use either “zero degree wet-bulb” height or “melting level height” here to avoid confusion.

Modified as: “Once the hailstones fall below the zero degree wet-bulb level they start to melt...”

4. Line 78: “responses to 60 dBZ” – do the authors mean responses of 60 dBZ storms to climate change? The authors should rephrase for clarity.

Modified as: “Another CPM study covering most of CONUS, found regional and seasonal differences in future changes in the number of storms reaching a reflectivity value of 60 dBZ (which could be indicative of hail; Haberlie et al. 2022), with mainly increases in the eastern U.S. and decreases over the Great Plains.”

5. Figure 1: The caption should specify the time period for the left column (I assume the historical period).

Modified accordingly: “Number of cases with daily maximum graupel water path (GWPmax) exceeding 10 kg m⁻² per year per km² (Graupel10) in the historical period, future (end-of-century) changes in Graupel10, and future (end-of-century) changes in Graupel50 (as in Graupel 10, but with 50 kg m⁻² threshold)...”

6. Lines 105-106: It would be good to also mention the large decreases over the ocean surrounding Italy in summer in this discussion.

Added a sentence to emphasize this point: “Large decreases are projected in the summer over the seas surrounding Italy (Fig 1h), despite the increases projected over Italy.”

7. Line 115: The definition of graupel should be in the previous section where graupel is first mentioned as a key feature of thunderstorms.

Moved the definition in parenthesis to the top of the previous accordingly: “Graupel (heavily rimed snow particles, of sizes up to 5 mm) production is a key feature of thunderstorms...”

8. Figure 2 left column: I am surprised that the severe hail potential shown by this proxy is not higher in the well-known hotspot of Northern Italy. Is the year-round hail potential really higher in northern Africa than in northern Italy? Although the proxy is explained in another paper (Kahraman et al. 2024) the authors should still explain these inconsistencies in their discussion of Figure 2 around lines 153-159. The authors can not claim that the proxy agrees with observations using only Supp. Fig 2 because that figure only covers Central Europe and not the key hail-prone regions in Southern Europe.

Thanks for pointing this out. Although our domain covers part of the northern African coastal zones, our focus in this study is Europe. We have excluded 70 grid points from the boundaries (white-masked within the map frames), but it can still be possible that the boundary effects might have played a role around further inner grids, as the simulation is basically a downscale from ERA-Interim’s 0.75 degree-grid to ~2.2km grid. This could include the mentioned African territories, perhaps more than eastern section of the domain. Our 20-y hindcast simulation (as seen on Fig 5 of Kahraman et al. 2024), and future simulation (Fig 2, right column show the difference between future and control, and most of these African storms disappear in the future) has less severe hail potential frequency over this particular region compared to northern Italy, so this might be specific to the control simulation, and would possibly be ironed out if -for instance- 30-y simulations could be achieved instead of 10-y. On the other hand, Prein and Holland (2018) show that this particular region indeed has very high values of large hail probability, even higher than northern Italy, based on their ERA-Interim-based proxy (their Fig 11). Given the region is unpopulated apart from the coast, it can’t be ruled out that this can be a hail zone without much report as well.

We have added the following to the lines aforementioned:

“Note that there are some interesting responses over northern Africa, but we do not consider these here due to their proximity to the lateral boundary of the model.”

Regarding the comment on Central Europe; actually our proxy at Kahraman et al. (2024) was compared with observations at other parts of Europe as well, albeit to a lesser extent. Figures 10a (number of severe hail reports across Europe in ESWD), and Figure 13 (ESWD peak month of severe hail) in that paper show wider European data. However, for clarity, we put only Central European data to our supplement here, and refer to the paper for further comparison. From Figure 10a in Kahraman et al. (2024), it is obvious that the observational database has made an enormous progress collecting the reports through the last two decades, and it is not really possible to treat all regions equally throughout the years, in terms of anticipating reporting efficiency. The huge difference in reporting practices among European regions is also discussed at Hulton and Schultz 2024, especially in the light of

data from Poland in mid 20th century (Fig 16 in their paper). For this reason, we have replaced the “only” to “mainly” in the sentence:

“...in Kahraman et al. (2024) we compared our proxy to observations mainly for Central Europe due to their reliability”

and cited the Hulton and Schultz (2024) reference.

One other thing to add here, is maybe the Eastern Adriatic coast, which features another hot zone except northern Italy. New studies (e.g. Blaskovic et al. 2023, Karaula 2024) spotlight this region’s hailstorms: their very high frequency (up to 120 hail days a year -although these include less than 2 cm hail), wintertime and early-morning peak, and decreasing multidecadal trend; which are perhaps relatively unexpected, but consistent with our findings.

9. Line 161: Which seasons are meant by “both seasons”?

This should have been summer. Modified as:

“We find that for warmer RCP8.5 end-of-century conditions, “severe hail potential” cases reduce by more than half in summer (Fig. 2, right column)...”

10. On the reliability of the proxy, Supp. Fig 2 shows that there is a larger difference between hail reports and proxy results in central Europe in Spring (MAM) than in other seasons, and one may question the reduction between present and future climate proxy-derived hail activity given that the future amount is actually more than the present reported amount.

MAM cases here are dominantly May cases (Fig 4e and Supp. Fig 7). There is evidence that the model is overestimating particularly in May, as shown in Kahraman et al. (2024). This could be reduced by adding a low-level moisture metric as a 5th component to the proxy, but then the storms in SON would much more increase in that case. In fact, 9 hail-relevant parameters were calculated and 4 were used in the proxy. This balanced solution with consistent definition covers the choice of minimum number of components with optimum estimation across seasons and regions. The reduction stems from the key components of the proxy.

European hail observations are imperfect, as the ESWD is a relatively new dataset, and relies on volunteer reports as well as a dedicated extraction from news, social media, etc. Hence, especially for periods else than the last decade, it is likely that only a small fraction of real events exist in the database, including the regions with the best coverage. Even for Central Europe, where the severe hail is likely better reported than elsewhere in our domain, it is obvious that underreporting is an issue in early years (Supp. Fig. 2b). 1999-2004 data indicate an increase from $\sim 10^1$ to $\sim 10^2$ events per year, which is ~ 5 -50 times less than recent years. Any normalization method for underreporting could increase the observation values more than what the model produced for all seasons. Nevertheless, we think that the difference in springtime results are noteworthy to be explicitly mentioned in the paper, and we have added the “except for higher values in spring” phrase to the statement:

“Overall, the seasonal distribution of severe hail potential in the control simulation agrees with the observations (Supp. Fig. 2) and the hindcast (Kahraman et al. 2024), except for higher values in spring.”

11. Supp. Fig 3. It is not possible for me to verify whether the changes shown here are significant or not without knowing the historical supercell frequency - the authors should either include historical supercell frequency in this plot or make the changes relative to historical values.

A comprehensive study on supercells doesn't exist in Europe, and this is something we are working on. We don't want to jeopardize the publication by putting more figures of supercell analysis here. However, in this paper, we use this plot only as a supplementary item, as our aim is not to show how European supercells are changing in the future, but to show that the reduction in the frequency of severe hail potential is not because there are essentially less supercells. Regarding the point we mention this supplementary figure:

"We find that for warmer RCP8.5 end-of-century conditions, "severe hail potential" cases reduce by more than half in summer (Fig. 2, right column), despite little change in supercell frequency, as estimated by an updraft helicity metric (Supp. Fig. 3a)."

We have now modified the figure to show the number of land grid points with Updraft Helicity values exceeding different thresholds for both control and future simulations in JJA. As we use the 50 m2/s2 threshold for supercells, this new figure shows that there is actually a small increase, rather than a decrease, which provides evidence to our claim: We clearly emphasize that severe hail potential cases are reduced, but there is little change in supercell frequency.

12. Line 166: "It is well known that once the liquid water content is sufficiently high, storms will more likely produce heavy rain resulting in flash floods, rather than severe hail." This statement requires at least one reference. The connection between this sentence and the results shown in this paragraph is also unclear - are the authors inferring an increase in liquid water content in their future scenario?

We have modified the sentence to better reflect the meaning, and added two references:

"It is well known that higher melting levels and larger warm cloud depths with high liquid water content will more likely produce heavy rain, rather than severe hail (e.g. Prein and Heymsfield 2020, Knight and Knight 2001)."

And thanks for the missing connection comment, now we have added the following sentence to make it clear that this is the case with our simulations:

"For instance, our future simulation has much higher melting level height (Kahraman et al. 2022), higher moisture availability (Kahraman et al. 2021), and more frequent MCSs (Chan et al. 2023) compared to the control simulation."

13. Lines 171-185: The annual results in Supp. Fig. 4 show a mid-century increase in severe hail potential around the Mediterranean Coast which then reverses and becomes a decrease by end of century. The authors should comment on this change in sign and the reasons for it.

We have added the following:

"Some localities near the Mediterranean coasts (southern France, offshore eastern Spain, and parts of Central Mediterranean Sea) show increasing frequency of severe hail potential in autumn in the mid-century simulation, but these diminish by end of century, mainly due to further reductions in summer (Supp. Fig. 4). This is likely due to the different evolution of competing mechanisms with time."

14. Line 185: Instead of stating that the significance of some results is “doubtful” in the text, the authors should actually test for statistical significance in the changes and show the results in Supp. Fig. 4 as they have for the results in main Fig. 2.

The significance test basically removes all low values around zero, and white out the north. As we show these at Fig 2 for end-of-century, we opt for presenting all data at this supplementary figure, in order to show how the values in Table 1 manifest themselves on the map. Nevertheless, we have removed the mentioned phrase to avoid speculation.

15. Figure 3: A significant downside to the analysis shown here is that the boundaries of the selected regions do not join each other. This means that for example the major hail region of the northern part of Switzerland and the Black Forest area in Germany are not covered by these analyses. The analysis of hail changes in Europe would be stronger if the regions were slightly expanded to include important hail zones like these. Otherwise, the authors should at least justify their choices of regions that exclude important hail zones.

As these are rare events, and the number of cases especially in the north is low, making regional analysis using grids with similar change signals is preferred here, rather than focusing on hail hotspots. The borders of these regions are mainly determined to indicate major geographical sections, keeping grid points with similar change signals together.

16. Lines 192-193: This analysis of the relative changes in HGZ updrafts vs. wind shear is valid for autumn but appears not to be true for the winter changes in severe hail potential (Supp. Fig. 5a and 5b). The authors should be careful with the wording here: I would suggest adding a sentence to deal separately with winter changes which appear to have to do with both wind shear than HGZ updraft strength changes.

Added a sentence to the end of the paragraph for winter:

“In winter, storms with severe hail potential tend to have both higher HGZ vertical velocities and higher deep layer shear in the future (Supp. Fig. 5).”

17. Lines 197-198: “supercells tend to occur in environments with high deep layer shear (expected to be the exclusive producers of significant severe hail) (Supp. Fig. 3).” Supp Fig. 3

shows the spatial distribution of supercells detected by updraft helicity so it does not support this statement about deep layer shear and whether deep layer shear environments are the exclusive producers of sig. severe hail. The statement requires a reference to support it.

Modified the sentence and added references accordingly:

“... supercells tend to occur in environments with high deep layer shear (e.g. Davies-Jones 2015), and are expected to be major producers of significant severe hail (e.g. Markowski and Richardson 2010, Allen et al. 2020, Wells et al. 2024).”

18. Table 2: The column “Change %” does not show relative change but rather the future climate amount as a percentage of the historic climate amount.

Thanks for pointing this out, the table and its caption is fixed. It has SigSH/SH ratios for current and future simulations, and the percentage changes. We have removed the word “relative” to avoid confusion.

19. Line 205: The authors should mention that the statement that Table 2 underestimates the shift to larger hail at the surface is hypothesised and therefore likely (I agree this is likely the case), but not shown by this work.

We have modified the statement accordingly:

“Hence, it is likely that the significant severe to severe hail potential ratio here (Table 2) represents a low estimate of the possible shift to larger hail at the surface.”

20. Figure 4e. This figure shows historical severe hail potential peaking in October in Europe.

Is this realistic? I would have expected a peak in the summer months and not in mid autumn. The authors should comment on the reliability of the proxy given this figure.

It is stated that the data of this figure includes whole domain, rather than land grids, at the end of the caption: “The analysis is for the whole domain analysed (i.e. excluding 70 grids from each lateral boundary).” From Fig 2, it is obvious that most of these storms appear over the sea, rather than land. The summer peak in Fig 4e indicates hailstorms over territorial area though. One can compare this with Supp. Fig. 9 (previously, this was Supp. Fig. 7), which shows land grid points only. This one clearly shows that current climate features June as the peak month of severe hail in Europe (as shown in observational studies; e.g. Pucik et al 2015, Hulton and Schultz 2024).

21. Lines 254-255: These statements about shifts in the peak month for severe hail are not backed up by Fig 4e which shows a peak in future severe hail potential in November. The result written here for sea points is not shown by either Fig 4e or Supp Fig 7.

Again, Fig 4 shows whole domain, and includes sea points. We modify our sentence to reflect it better (note that this is in the “Variability and drivers for changes” section, later in the manuscript, and the Supp. Fig. 7 is now Supp. Fig. 9):

“There is a notable shift of the peak month from June towards May over land (Supp. Fig 9) and from October to December over sea.”

22. Line 255: The reason given for the decreases in warm season variability – that they “stem from warming” is too vague. What is it about warming that explains the less frequent strong updrafts? A thermodynamic interpretation would lead to expectations of increased updraft strengths with warming (as the authors note later).

We have modified the sentence as following:

“Decreases in warm season variability stem from a projected shift in thunderstorms towards warmer FZLV ranges.”

23. Supp. Figures 8 and 9: Both these figures need to show the statistical significance of the observed changes to a given level of significance.

The EXP2 and EXP3 figures are now removed, and replaced with new analysis. EXP1 figure (Supp Fig. 3) was renewed, including application of a mask for insignificant changes (in white).

24. Line 291: Here the authors ascribe their contrasting results partly to previous studies not including freezing level height changes, yet the authors also say in the previous section that the reductions they observe in storm occurrences occur without freezing level changes included.

Here we aim to analyse the reasons for the differences. We don't analyse the mentioned entrainment etc. features as well, but these are treated within the model, and the methods could be a factor on the contrasting results. For instance, if the freezing level changes have been used in the cited references, it can well be argued that a (relative) reduction in hailstorms could be found, or at least, the amount of increases could be reduced. This is independent of our results showing still decreases with or without FZLV. Nevertheless, we have modified our sentence and moved the location of the freezing level height statement in the sentence:

“Differences, at least partly, stem from the study methodologies. In particular, previous studies do not include the effects of warming on HGZ level changes, freezing level height changes, entrainment, precipitation loading, etc., that counteract the effects of increases in instability; previous studies solely focus on environmental conditions rather than simulated convection, thus increases in instability with atmospheric warming dominate their results.”

25. Line 606: Thanks to the authors' response, I now understand why hail information from the convective-permitting model was not used (ie it wasn't produced). The authors should note explicitly in the methods section what hail information was and was not available in the UM outputs.

We have added the relevant information in the methods section:

“An explicit hail parameter doesn't exist in the stored model output, but graupel does.”

26. Line 611: I notice that the authors never write the date range for the future scenario, instead they write “ca. 2100” or similar. What was the date range of the boundary conditions used?

The end-of-century simulation represents a global warming level (under RCP8.5) corresponding to the year 2100, per se boundary conditions coming from the GCM.

27. Line 610: I previously noted that 10 years is a small length of time for per-pixel comparisons, given the high interannual variability for storms. The authors responded that they agree and that the results are spatially aggregated. More detail needs to be shown about how this is done. It appears the authors are referring to smoothing of the observed changes, across space, but it is unclear whether the smoothing is applied before or after the comparison of the timeseries and significance testing.

The smoothing is done for the control and future data separately, and significance testing is done after the comparison. We now describe it in the methods:

“Severe hail is a rare hazard for a locality, so an analysis in a 2km grid, with 10-year worth of data is very noisy. Hence, we apply a spatial smoothing technique, visualising frequencies per 10,000 km² (i.e. per 100 km × 100 km). This is done by averaging the neighbouring ± 25 grid points in each direction. “

28. Line 620: Here, the authors should acknowledge that the assumption that “all convection is explicitly resolved with a 2.2 km grid” is strong, given what is known about the resolutions required to properly model deep convection (e.g. Bryan et al., 2003).

Fully agreed. Now the sentence reads as the following:

“No cumulus parameterization is set, as all convection is assumed to be explicitly resolved with a 2.2km grid.”

29. Line 607: The first sentence is missing a word (“were run?”).

Added the word “were”.

References

Blašković L., Jelić D., Malešić B., Omazić B., Güttler I., Telišman Prtenjak M. (2023.): Trend analysis and climatology of hail in Croatia. *Atmospheric Research*, 294, 106927. <https://doi.org/10.1016/j.atmosres.2023.106927>

Bryan, G. H., J. C. Wyngaard, and J. M. Fritsch, 2003: Resolution requirements for the simulation of deep moist convection. *Monthly Weather Review*, 131 (10), 2394–2416, doi: 10.1175/1520-0493(2003)131<2394:RRFTSO>2.0.CO;2.

Chan, S.C., Kendon, E.J., Fowler, H.J. et al. Large-scale dynamics moderate impact-relevant changes to organised convective storms. *Commun Earth Environ* 4, 8 (2023). <https://doi.org/10.1038/s43247-022-00669-2>

Davies-Jones, R. (2015), A review of supercell and tornado dynamics, *Atmospheric Research*, 158–159, 274-291, <https://doi.org/10.1016/j.atmosres.2014.04.007>.

Hulton, F. and Schultz, D. M.: Climatology of large hail in Europe: characteristics of the European Severe Weather Database, *Nat. Hazards Earth Syst. Sci.*, 24, 1079–1098, <https://doi.org/10.5194/nhess-24-1079-2024>, 2024.

Kahraman A, Kendon EJ, Chan SC, Fowler HJ (2021). Quasi-stationary intense rainstorms spread across Europe under climate change. *Geoph. Res. Lett.* 48, e2020GL092361. <https://doi.org/10.1029/2020GL092361>

Kahraman A, Kendon EJ, Fowler HJ, Wilkinson J., (2022). Contrasting future lightning stories across Europe. *Env. Res. Lett.*, 17(11), 114023.

Karaula, Lovro, 2024: Early morning hail over the southern part of the Adriatic. Graduation thesis, <https://repozitorij.pmf.unizg.hr/islandora/object/pmf:13732>

Kim, M.H.; Lee, J.; Lee, S.-J. Hail: Mechanisms, Monitoring, Forecasting, Damages, Financial Compensation Systems, and Prevention. *Atmosphere* 2023, 14, 1642. <https://doi.org/10.3390/atmos14111642>

Knight and , C. A., and N. C. Knight, 2001: Hailstorms. *Meteor. Monogr.*, 28, 223–248, <https://doi.org/10.1175/0065-9401-28.50.223>.

Lamb, D., and J. Verlinde, 2011: Chapter 8.4: Melting. *Physics and Chemistry of Clouds*, Cambridge University Press.

Martín, M. L., Calvo-Sancho, C., Taszarek, M., González-Alemán, J. J., Montoro-Mendoza, A., Díaz-Fernández, J., et al. (2024). Major role of marine heatwave and anthropogenic climate change on a Giant hail Event in Spain. *Geophysical Research Letters*, 51, e2023GL107632. <https://doi.org/10.1029/2023GL107632>

Prein and Holland (2018) Global estimates of damaging hail hazard. *Weather Clim Extremes* 22:10–23. <https://doi.org/10.1016/j.wace.2018.10.004>

Prein, A. F., Wang, D., Ge, M., Ramos Valle, A., & Chasteen, M. B. (2025). Resolving mesoscale convective systems: Grid spacing sensitivity in the tropics and midlatitudes. *Journal of Geophysical Research: Atmospheres*, 130, e2024JD042530. <https://doi.org/10.1029/2024JD042530>

Sari, F. P., & Lasher-Trapp, S. (2025). Hailstorm events over a maritime tropical region: Storm environments and characteristics. *Journal of Geophysical Research: Atmospheres*, 130, e2024JD042718. <https://doi.org/10.1029/2024JD042718>

Taszarek, M., J. T. Allen, T. Půcik, K. A. Hoogewind, and H. E. Brooks, 2020: Severe convective storms across europe and the united states. part ii: Era5 environments associated with lightning, large hail, severe wind, and tornadoes. *J Climate*, 33 (23), 10 263 – 10 286, doi: 10.1175/JCLI-D-20-0346.1.

Wells HM, Hillier J, Garry FK, Dunstone N, Clark MR, Kahraman A, Chen H, 2024. Climatology and convective mode of severe hail in the United Kingdom. *Atmos. Res.* 309: 107569. <https://doi.org/10.1016/j.atmosres.2024.107569>

Yeo, S., R. Leigh, and I. Kuhne, 1999: The April 1999 Sydney Hailstorm. *Aust. J. Emergency Manage.*, 14 (4), 23–25.

Response to Review

Review of NCOMMS-24-10209B

This re-review is for manuscript NCOMMS-24-10209B, titled "Future changes in severe hail across Europe moderated by new warm thunderstorm type".

I thank the authors for their thorough responses to my review comments. In particular I thank the authors for their very thorough and thoughtful response to my comments around the drivers of the changes. The analysis is significantly improved.

I'm now satisfied that the authors have updated the manuscript sufficiently and addressed all my concerns, and with the exception of a few minor comments below I now believe the manuscript is ready for acceptance.

I wish the authors all the best for this interesting and thorough work.

We thank the reviewer for their constructive reviews during the process, which helped improving the paper substantially.

We note that the title of the manuscript is now "Future changes in severe hail across Europe including regional emergence of warm-type thunderstorms", addressing the editor's comment.

1. Line 24: "Using the first pan-European convection-permitting simulations" – but on lines 88-90 the authors say that apart from their study there are "no previous pan-continental CPM studies for Europe studying future changes in (severe) hail using a physically-based hail diagnostic". Are these the first pan-European convection-permitting simulations ever produced, or is this the first study to use pan-European CPMs to study severe hail?

We have removed the "using the first" wording from the abstract (replaced with "Here we use...").

For clarity, we have also rewritten the mentioned sentence (lines 89-90) as:

"To date, there have been no pan-continental CPM studies for Europe that examine projected changes in severe hail using a physically-based hail diagnostic approach."

2. Lines 176–179: "For instance, our future simulation has much higher melting level height (Kahraman et al. 2022), higher moisture availability (Kahraman et al. 2021), and more frequent MCSs (Chan et al. 2023) compared to the control simulation." The authors say here that their simulations show these changes but do not show them in this manuscript and refer to other authors. How are more frequent MCSs shown in these simulations?

The cited references all make use of the same simulation suite, and are produced by our group. To make it clear, we have modified this sentence as:

“For instance, as shown by earlier studies, our future simulation has much higher melting level height ⁽³⁰⁾, higher moisture availability ⁽³⁵⁾, and more frequent MCSs (as shown by applying a precipitation tracking algorithm³⁶) compared to the control simulation.”

3. Lines 211–214: The results stemming from Supplementary Figure 5 are not all that convincing and I think require at least an indication of statistical significance which is difficult to estimate from the figure.

We appreciate the reviewer’s observation regarding Supplementary Figure 5. We would like to clarify that significant severe hail potential pertains to a very rare phenomenon, which inherently limits the statistical power and the applicability of conventional significance testing. Due to the low frequency of occurrence, standard statistical tests may not yield meaningful or reliable p-values, and could potentially misrepresent the underlying signal.

Nonetheless, we have revised the figure legend to explicitly acknowledge this limitation to clarify why statistical significance was not applied in this context. We hope this additional context helps convey the interpretive boundaries of the data more clearly.

4. Lines 285–286: “and from October to December over sea” – add “(not shown)” here?

Added.

Review of NCOMMS-24-10209A

This review is for manuscript NCOMMS-24-10209A, titled “Future changes in large hail across Europe dominated by new warm thunderstorm type”.

The study shows an evaluation of proxy-derived changes in frequency of severe hail conditions across Europe. The proxy uses features derived from a convection-permitting model rather than broader environmental analysis, yet it remains a proxy with the well-known disconnect between hail-prone conditions and hail occurrence. The results show decreases in severe hail potential across Europe, particularly in the warm season, with some cool season increases. The paper identifies that more tropical-like storms with high melting levels occur in the future scenario, but the claim that they dominate changes in large hail is untested.

The paper is an application of an innovative proxy across a large domain and the results will help to untangle the complex effects of climate change on hailstorms. It is excellent that Kahraman et al. (2024) has been published, providing justification for and detailed analysis of the proxy used in this study.

While the results of the study are useful and interesting there remain serious shortcomings with the paper that mean that, unfortunately, it requires further revision before being acceptable for publication. I hope that my suggested changes below allow for the manuscript to be improved.

The biggest issues are as follows:

1. **Analysis of change drivers:** In a previous review of this manuscript I wrote that the authors’ method for analysing the drivers of changes was flawed. The authors’ responded that running new simulations is too computationally expensive, which I acknowledge is a difficulty (other approaches such as removal of biases in ingredients could be considered). The problem of unfounded claims regarding the drivers of changes remains in the revised manuscript.

In particular, on lines 260-275, the results (EXP1 and EXP2) are interesting but they still do not show what the authors claim in the text. As the authors’ acknowledged in their responses, as soon as the proxy is changed by removing threshold(s), the proxy is no longer detecting the same storms but rather a larger set of storms, including those that were not detected because of the thresholds in the original proxy. While changes in this larger set may imply driving factors, as the authors wrote in their responses, any changes do not necessarily explain changes in the original set. As I wrote in my previous review, to properly determine the driving factors for the observed changes, the authors would need to run the same proxy

on simulations in which changes to certain aspects (e.g. FLZ) had been removed. Alternatively, the authors could use their existing data and look at changes in individual proxy ingredients for the set of storms identified using the original proxy to see which ingredients had the largest changes (this too is imperfect since the largest changes may not be the most explanatory).

There is useful information to be found by comparing changes using different proxies, as the authors have done, but care must be taken in how those changes are described. This is because the modified proxies may no longer detect only severe hail conditions and therefore cannot be used to show the drivers for changes in severe hail. For example the sentence on line 268 “We find that a decrease in the occurrence of high vertical velocities in the HGZ is the dominant factor leading to a decrease in severe hail” is not backed up by the evidence shown in the paper. On the other hand, an earlier sentence starting on line 262 “On removal of the freezing level height parameter from the proxy, future decreases are still seen for all seasons (Supp. Fig. 8), which implies that future thunderstorms with high shear and high vertical velocities in the HGZ are less frequent” is fine.

2. **The new thunderstorm type:** I wrote previously that more detail is required about the new “warm thunderstorm” type that the authors describe in this work. To their credit the authors have added significant detail with references to the literature, and more good analyses. Figure 5 does a good job of showing the increases in these “warm type” thunderstorms. What remains missing is justification for the title of the paper which says that changes in hail are dominated by this new type. To justify that headline claim, the authors need to show at least the proportion of the changes that are related to the warm thunderstorm type, and that they outweigh other changes.

The main basis for the new type is the bimodal feature in freezing level heights for thunderstorms with high graupel content in autumn – so it is not necessarily for hail-bearing thunderstorms and it is for only one season (the summer results appear marginal). It is unclear to me why the authors have not shown the connection by testing whether this bimodal distribution also appears for times that the proxy indicates are prone to severe or significant severe hail.

3. **Proxy vs. actual hail:** Care needs to be taken across the whole manuscript that the results are not misstated as showing changes in actual hail occurrence. Rather they are showing changes in proxy-derived hail potential. As an example, the abstract starts well but then mentions the “decreases in severe hail”, when no such decreases are shown and what is shown is decreases in severe hail potential. This disconnect between hail-prone conditions and hail occurrence exists for all proxy studies, and as I noted previously, most hail-prone environments do not result in hail production (e.g. Taszarek et al., 2020, for Europe).
4. **The mid-century scenario:** In the text the authors consistently refer to a date range of 1940-1949 as a “future climate”, which is very confusing. Even in the methods section the date range is given as 1940-1949. Supp. Fig 4 subplot titles indicate that the period in question is actually 2040-2049 which would make more sense. I have read the paper assuming that when

the authors wrote 1940-1949 they actually meant 2040-2049. Although this is essentially a consistent typo, it added to confusion reading the paper and obviously needs clarification and fixing.

Specific comments

1. Line 32: The results “suggest we should be prepared for (infrequent but) much larger hail in a future warmer world.” Yet the paper’s analysis of hail size is restricted to changes in the ratio of sig. severe to severe hail environments. There is no analysis of actual hail size in the results. My opinion is that this line is overselling the results.
2. Line 42: I maintain that a reference on the challenges of short-term hail forecasting is required here. Allen 2020 does deal with forecasting while Raupach 2021 does not. The sentence should read “there remain uncertainties about how it forms (Allen et al. 2020), challenges around its short-range forecasting (**reference**) and its projection with a changing climate (Raupach et al. 2021)”.
3. Line 63: As discussed in my previous review hailstone melting can be delayed to well below the zero degree dry-bulb level (e.g. Lamb and Verlinde, 2011) and the authors should use either “zero degree wet-bulb” height or “melting level height” here to avoid confusion.
4. Line 78: “responses to 60 dBZ” – do the authors mean responses of 60 dBZ storms to climate change? The authors should rephrase for clarity.
5. Figure 1: The caption should specify the time period for the left column (I assume the historical period).
6. Lines 105-106: It would be good to also mention the large decreases over the ocean surrounding Italy in summer in this discussion.
7. Line 115: The definition of graupel should be in the previous section where graupel is first mentioned as a key feature of thunderstorms.
8. Figure 2 left column: I am surprised that the severe hail potential shown by this proxy is not higher in the well-known hotspot of Northern Italy. Is the year-round hail potential really higher in northern Africa than in northern Italy? Although the proxy is explained in another paper (Kahraman et al. 2024) the authors should still explain these inconsistencies in their discussion of Figure 2 around lines 153-159. The authors can not claim that the proxy agrees with observations using only Supp. Fig 2 because that figure only covers Central Europe and not the key hail-prone regions in Southern Europe.
9. Line 161: Which seasons are meant by “both seasons”?
10. On the reliability of the proxy, Supp. Fig 2 shows that there is a larger difference between hail reports and proxy results in central Europe in Spring (MAM) than in other seasons,

and one may question the reduction between present and future climate proxy-derived hail activity given that the future amount is actually more than the present reported amount.

11. Supp. Fig 3. It is not possible for me to verify whether the changes shown here are significant or not without knowing the historical supercell frequency - the authors should either include historical supercell frequency in this plot or make the changes relative to historical values.
12. Line 166: “It is well known that once the liquid water content is sufficiently high, storms will more likely produce heavy rain resulting in flash floods, rather than severe hail.” This statement requires at least one reference. The connection between this sentence and the results shown in this paragraph is also unclear - are the authors inferring an increase in liquid water content in their future scenario?
13. Lines 171-185: The annual results in Supp. Fig. 4 show a mid-century increase in severe hail potential around the Mediterranean Coast which then reverses and becomes a decrease by end of century. The authors should comment on this change in sign and the reasons for it.
14. Line 185: Instead of stating that the significance of some results is “doubtful” in the text, the authors should actually test for statistical significance in the changes and show the results in Supp. Fig. 4 as they have for the results in main Fig. 2.
15. Figure 3: A significant downside to the analysis shown here is that the boundaries of the selected regions do not join each other. This means that for example the major hail region of the northern part of Switzerland and the Black Forest area in Germany are not covered by these analyses. The analysis of hail changes in Europe would be stronger if the regions were slightly expanded to include important hail zones like these. Otherwise, the authors should at least justify their choices of regions that exclude important hail zones.
16. Lines 192-193: This analysis of the relative changes in HGZ updrafts vs. wind shear is valid for autumn but appears not to be true for the winter changes in severe hail potential (Supp. Fig. 5a and 5b). The authors should be careful with the wording here: I would suggest adding a sentence to deal separately with winter changes which appear to have to do with both wind shear than HGZ updraft strength changes.
17. Lines 197-198: “supercells tend to occur in environments with high deep layer shear (expected to be the exclusive producers of significant severe hail) (Supp. Fig. 3).” Supp Fig. 3 shows the spatial distribution of supercells detected by updraft helicity so it does not support this statement about deep layer shear and whether deep layer shear environments are the exclusive producers of sig. severe hail. The statement requires a reference to support it.
18. Table 2: The column “Change %” does not show relative change but rather the future climate amount as a percentage of the historic climate amount.
19. Line 205: The authors should mention that the statement that Table 2 underestimates the shift to larger hail at the surface is hypothesised and therefore likely (I agree this is likely the case), but not shown by this work.

20. Figure 4e. This figure shows historical severe hail potential peaking in October in Europe. Is this realistic? I would have expected a peak in the summer months and not in mid autumn. The authors should comment on the reliability of the proxy given this figure.
21. Lines 254-255: These statements about shifts in the peak month for severe hail are not backed up by Fig 4e which shows a peak in future severe hail potential in November. The result written here for sea points is not shown by either Fig 4e or Supp Fig 7.
22. Line 255: The reason given for the decreases in warm season variability – that they “stem from warming” is too vague. What is it about warming that explains the less frequent strong updrafts? A thermodynamic interpretation would lead to expectations of increased updraft strengths with warming (as the authors note later).
23. Supp. Figures 8 and 9: Both these figures need to show the statistical significance of the observed changes to a given level of significance.
24. Line 291: Here the authors ascribe their contrasting results partly to previous studies not including freezing level height changes, yet the authors also say in the previous section that the reductions they observe in storm occurrences occur without freezing level changes included.
25. Line 606: Thanks to the authors’ response, I now understand why hail information from the convective-permitting model was not used (ie it wasn’t produced). The authors should note explicitly in the methods section what hail information was and was not available in the UM outputs.
26. Line 611: I notice that the authors never write the date range for the future scenario, instead they write “ca. 2100” or similar. What was the date range of the boundary conditions used?
27. Line 610: I previously noted that 10 years is a small length of time for per-pixel comparisons, given the high interannual variability for storms. The authors responded that they agree and that the results are spatially aggregated. More detail needs to be shown about how this is done. It appears the authors are referring to smoothing of the observed changes, across space, but it is unclear whether the smoothing is applied before or after the comparison of the timeseries and significance testing.
28. Line 620: Here, the authors should acknowledge that the assumption that “all convection is explicitly resolved with a 2.2 km grid” is strong, given what is known about the resolutions required to properly model deep convection (e.g. Bryan et al., 2003).
29. Line 607: The first sentence is missing a word (“were run”?).

References

- Bryan, G. H., J. C. Wyngaard, and J. M. Fritsch, 2003: Resolution requirements for the simulation of deep moist convection. *Monthly Weather Review*, **131** (10), 2394–2416, doi: 10.1175/1520-0493(2003)131<2394:RRFTSO>2.0.CO;2.

Lamb, D., and J. Verlinde, 2011: Chapter 8.4: Melting. *Physics and Chemistry of Clouds*, Cambridge University Press.

Taszarek, M., J. T. Allen, T. Púčik, K. A. Hoogewind, and H. E. Brooks, 2020: Severe convective storms across europe and the united states. part ii: Era5 environments associated with lightning, large hail, severe wind, and tornadoes. *J Climate*, **33** (23), 10 263 – 10 286, doi: 10.1175/JCLI-D-20-0346.1.

Review of NCOMMS-24-10209B

This re-review is for manuscript NCOMMS-24-10209B, titled “Future changes in severe hail across Europe moderated by new warm thunderstorm type”.

I thank the authors for their thorough responses to my review comments. In particular I thank the authors for their very thorough and thoughtful response to my comments around the drivers of the changes. The analysis is significantly improved.

I’m now satisfied that the authors have updated the manuscript sufficiently and addressed all my concerns, and with the exception of a few minor comments below I now believe the manuscript is ready for acceptance.

I wish the authors all the best for this interesting and thorough work.

1. Line 24: “Using the first pan-European convection-permitting simulations” – but on lines 88-90 the authors say that apart from their study there are “no previous pan-continental CPM studies for Europe studying future changes in (severe) hail using a physically-based hail diagnostic”. Are these the first pan-European convection-permitting simulations ever produced, or is this the first study to use pan-European CPMs to study severe hail?
2. Lines 176–179: “For instance, our future simulation has much higher melting level height (Kahraman et al. 2022), higher moisture availability (Kahraman et al. 2021), and more frequent MCSs (Chan et al. 2023) compared to the control simulation.” The authors say here that their simulations show these changes but do not show them in this manuscript and refer to other authors. How are more frequent MCSs shown in these simulations?
3. Lines 211–214: The results stemming from Supplementary Figure 5 are not all that convincing and I think require at least an indication of statistical significance which is difficult to estimate from the figure.
4. Lines 285–286: “and from October to December over sea” – add “(not shown)” here?